# Preparation for upcoming attentional states in the hippocampus and medial prefrontal cortex

**Eren Günseli[1,2]\*, Mariam Aly[1,3]**

[1]Department of Psychology, Columbia University, New York, United States; [2]Department of Psychology, Sabanci University, Istanbul, Turkey; [3]Affiliate Member, Zuckerman Mind Brain Behavior Institute, Columbia University, New York, United States

**Abstract** Goal-directed attention is usually studied by providing individuals with explicit instructions on what they should attend to. But in daily life, we often use past experiences to guide our attentional states. Given the importance of memory for predicting upcoming events, we hypothesized that memory-guided attention is supported by neural preparation for anticipated attentional states. We examined preparatory coding in the human hippocampus and mPFC, two regions that are important for memory-guided behaviors, in two tasks: one where attention was guided by memory and another in which attention was explicitly instructed. Hippocampus and mPFC exhibited higher activity for memory-guided vs. explicitly instructed attention. Furthermore, representations in both regions contained information about upcoming attentional states. In the hippocampus, this preparation was stronger for memory-guided attention, and occurred alongside stronger coupling with visual cortex during attentional guidance. These results highlight the mechanisms by which memories are used to prepare for upcoming attentional goals.

**\*For correspondence:**
gunseli.eren@gmail.com

**Competing interests:** The authors declare that no competing interests exist.

## Introduction

Humans continuously experience rich perceptual input — input that exceeds the brain's information processing capacity (*Luck and Vogel, 1997*; *Pylyshyn and Storm, 1988*; *Raymond et al., 1992*). As a result, only a small portion of the information that is encountered on a moment-by-moment basis is fully processed. Indeed, unless attended, even very salient information can go undetected (*Neisser and Becklen, 1975*; *Simons and Chabris, 1999*). Despite this severe limitation in information processing capacity, we can adaptively and efficiently function in the complex environment around us. How do we figure out what to attend and what to ignore in the face of rich, multidimensional input?

In laboratory studies, goal-directed attention is typically studied by providing explicit instructions to participants (*Posner, 1980*; *Kastner and Ungerleider, 2000*; *Wolfe et al., 1989*). For example, in cued attention tasks, participants are given particular target images or object categories that should be attended and detected (e.g., 'find a human in this picture'; *Wolfe et al., 2011*). These studies have very compellingly shown that humans can guide attention based on top-down goals and highlighted the neural mechanisms that allow this to happen (*Gazzaley and Nobre, 2012*; *Hopfinger et al., 2000*). However, in daily life, it is exceedingly rare to receive explicit instructions on how we should direct our attention. Instead, our attentional states are often guided by past experiences in similar situations (*Awh et al., 2012*). Such *memory-guided attention* is effective in guiding goal-directed behavior (*Aly and Turk-Browne, 2017*; *Chen and Hutchinson, 2018*; *Nobre and Stokes, 2019*) but is relatively under-explored. Here, we examine the mechanisms underlying memory-guided attention with the aim of determining the nature of neural representations that enable

**eLife digest** At any given moment, humans are bombarded with a constant stream of new information. But the brain can take in only a fraction of that information at once. So how does the brain decide what to pay attention to and what to ignore? Many laboratory studies of attention avoid this issue by simply telling participants what to attend to. But in daily life, people rarely receive instructions like that. Instead people must often rely on past experiences to guide their attention. When cycling close to home, for example, a person knows to watch out for the blind junction at the top of the hill and for the large pothole just around the corner.

Günseli and Aly set out to bridge the gap between laboratory studies of attention and real-world experience by asking healthy volunteers to perform two versions of a task while lying inside a brain scanner. The task involved looking at pictures of rooms with different shapes. Each room also contained a different painting. In one version of the task, the volunteers were told to pay attention to either the paintings or to the room shapes. In the other version, the volunteers had to use previously memorized cues to work out for themselves whether they should focus on the paintings or on the shapes.

The brain scans showed that two areas of the brain with roles in memory – the hippocampus and the prefrontal cortex – were involved in the task. Notably, both areas increased their activity when the volunteers used memory to guide their attention, compared to when they received instructions telling them what to focus on. Moreover, patterns of activity within the hippocampus and prefrontal cortex contained information about what the participants were about to focus on next – even before volunteers saw the particular picture that they were supposed to pay attention to. In the hippocampus, this was particularly the case when the volunteers based their decisions on memory.

These results reveal a key way in which humans leverage memories of past experiences to help optimize future behavior. Understanding this process could shed light on why memory impairments make it harder for people to adjust their behavior to achieve specific goals.

past experiences to be used to prepare for upcoming attentional states. We define 'attentional state' as the prioritized processing of particular environmental features in order to perform a given task. This entails focusing on task-relevant features, often at the expense of task-irrelevant features. Attentional states can be considered an instance of a task representation or a task set (*Mayr and Kliegl, 2000*; *Sakai, 2008*), with the task defining what should be attended to.

What brain regions may establish memory-guided attentional states? We focus on two candidate regions, the hippocampus and medial prefrontal cortex (mPFC). Interactions between these regions have been linked to a variety of goal-directed behaviors that are guided by long-term memory (*Euston et al., 2012*; *Kaplan et al., 2017*; *Shin and Jadhav, 2016*). Furthermore, both the hippocampus (*Aly and Turk-Browne, 2016a*; *Aly and Turk-Browne, 2016b*; *Aly and Turk-Browne, 2018*; *Córdova et al., 2019*; *Fenton et al., 2010*; *Mack et al., 2016*; *Muzzio et al., 2009*; *Ruiz et al., 2020*) and mPFC (*Mack et al., 2016*; *Small et al., 2003*) contribute to attentional processing. These findings suggest that the hippocampus and mPFC may work together to guide attentional behaviors on the basis of memory. Below, we explore their potential roles in more detail.

Previous work from our lab has demonstrated that the hippocampus represents online attentional states (*Aly and Turk-Browne, 2016a*; *Aly and Turk-Browne, 2016b*; *Córdova et al., 2019*). Moreover, decades of work have highlighted the critical role of the hippocampus in encoding and retrieving long-term memories (*Lepage et al., 1998*; *Shapiro and Eichenbaum, 1999*). These findings therefore suggest that the hippocampus might play an important role in establishing memory-guided attentional states. In line with this, several studies have found that hippocampal activity levels are higher for memory-guided vs. explicitly instructed attention (*Aly and Turk-Browne, 2017*; *Stokes et al., 2012*; *Summerfield et al., 2006*). This activity enhancement for memory-guided attention is present as soon as information from memory is available, and even prior to attentional guidance. This suggests that the hippocampus may be using memory to direct attentional states in a *preparatory* fashion: Hippocampal memories might prepare perception for attentional requirements that are anticipated based on previous experiences (*Stokes et al., 2012*).

However, enhanced activity levels are ambiguous and do not by themselves establish what a brain region is doing to guide attention on the basis of memory. One possibility is that the hippocampus simply retrieves a memory that is then used by other brain areas to guide attention. An alternative possibility is that the hippocampus is itself engaged in the process of guiding attention based on past experience. For example, when using past experience to anticipate a navigational goal on the right-hand side, it could be that (1) the hippocampus retrieves a memory that your desired location is on the right, and other brain areas use that information to guide attention; or (2) the hippocampus itself codes for a rightward attentional bias in preparation for detecting the navigational goal. Because our prior studies have indicated that the hippocampus can represent attentional states that are currently in play (*Aly and Turk-Browne, 2016a*; *Aly and Turk-Browne, 2016b*; *Córdova et al., 2019*) we hypothesized that it can also represent attentional goals that are retrieved from memory, and use those to prepare for upcoming attentional tasks (*Stokes et al., 2012*; *Summerfield et al., 2006*).

Beyond the hippocampus, mPFC may play an important role in memory-guided attention. In rodents, increased neural synchrony between the hippocampus and mPFC has been observed at decision points in which memory must be used to guide future behavior (*Benchenane et al., 2010*; *Jones and Wilson, 2005*). In humans, functional magnetic resonance imaging (fMRI) studies have demonstrated that the orbitofrontal cortex (a region in the ventral medial prefrontal cortex) represents goal state representations that are not explicitly instructed but rather inferred on the basis of past experience (*Niv, 2019*; *Schuck et al., 2015*; *Schuck et al., 2016*). Moreover, the hippocampus and ventromedial PFC (vmPFC) show functional coupling as individuals learn which features of an object are relevant for determining its category, and thus should be attended (*Mack et al., 2016*). Based on these studies, we predicted that vmPFC might also represent memory-guided attentional states.

To test if the hippocampus and vmPFC represent attentional states that are guided by memory, we used a novel behavioral task in conjunction with representational similarity analyses (*Kriegeskorte et al., 2008*). We were inspired by past work that demonstrated enhanced hippocampal activity in anticipation of attentional goals that were known based on memory (*Stokes et al., 2012*) as well as findings that link enhanced vmPFC activity to behavioral benefits that are attributed to the preparatory allocation of attention (*Small et al., 2003*). Based on this work and the other findings noted above, we predicted that the hippocampus and vmPFC will establish memory-based attentional states prior to when those states must be used. To this end, we first sought to determine whether these regions can differentiate between different online attentional states, and then tested whether neural signatures of these states can be detected prior to the attentional task itself — with the hypothesis that these regions will prepare for upcoming attentional states primarily when they are guided by memory.

We therefore compared attention in two tasks: One where attention was explicitly instructed, and one where attention was guided by memory. These tasks were modifications of ones we have previously used to demonstrate hippocampal representations of online attentional states (*Aly and Turk-Browne, 2016a*; *Aly and Turk-Browne, 2016b*). One key feature of these tasks is that they require relational representations, which are known to be strong drivers of hippocampal function (*Aly et al., 2013*; *Aly and Turk-Browne, 2018*; *Brown and Aggleton, 2001*; *Cohen and Eichenbaum, 1993*; *Davachi, 2006*; *Hannula and Ranganath, 2008*).

Participants were shown sequentially presented images of 3D-rendered rooms, each of which had several pieces of furniture, unique configurations of wall angles, and a single painting (*Figure 1*). In the *explicitly-instructed* task, participants received a cue prior to the first image (the base image) that told them to pay attention to either the style of the paintings ('ART') or the spatial layout of the rooms ('ROOM'). Following the base image, participants viewed a search set of 4 other images. On 'art' trials, they were to attend to the style of the paintings, and indicate whether any of the paintings in the search set could have been painted by the same person who painted the painting in the base image. On 'room' trials, they were to attend to the layout of the rooms, and indicate whether any of the rooms in the search set had the same spatial layout as the base image, but viewed from a slightly different perspective. Finally, participants received a probe ('ART?' or 'ROOM'?) and had to indicate if any of the search images matched the base image in the probed category (i.e., painting by the same artist, or room with the same spatial layout).

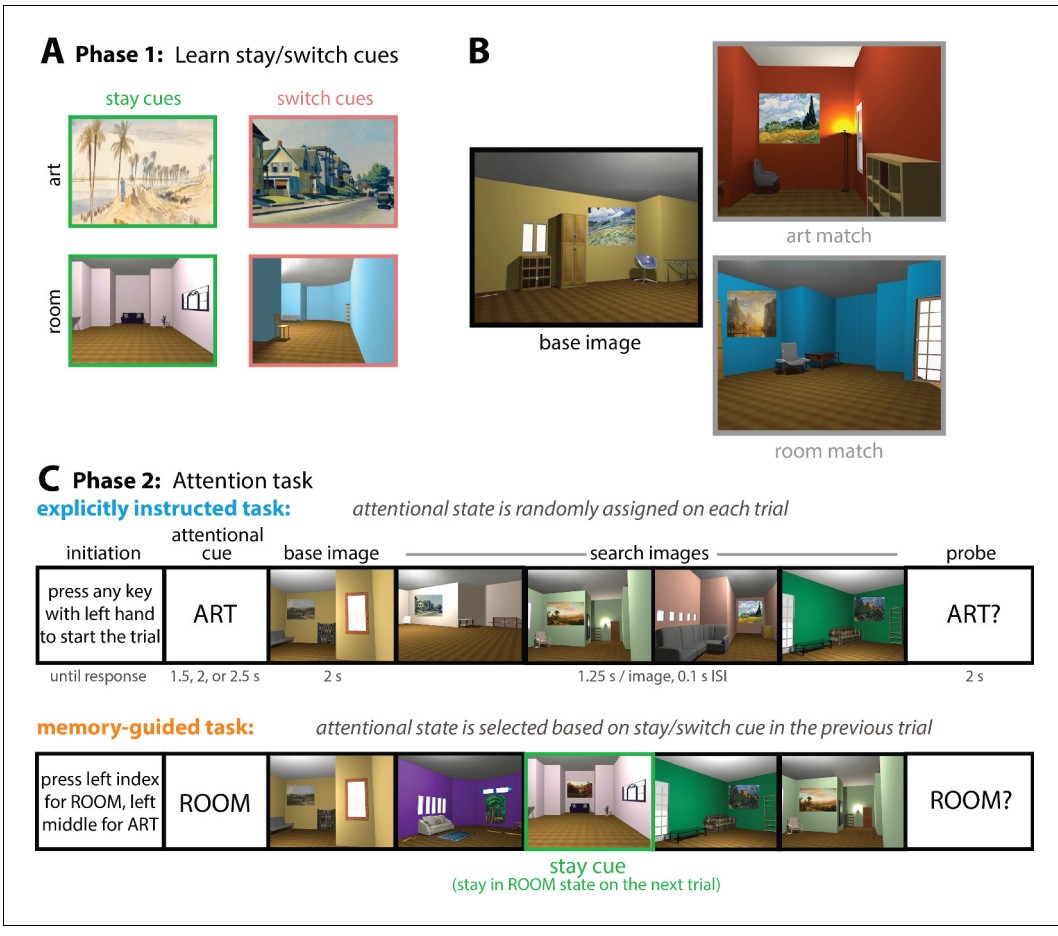

**Figure 1.** Task design. (**A**) Before entering the MRI scanner, participants learned stay and switch cues (Phase 1) that would be embedded in a subsequent attention task (Phase 2). One painting and one room were 'stay' cues, and one painting and one room were 'switch' cues. 'Stay' cues indicated that, during the subsequent memory-guided attention task, participants should stay in the same attentional state on the following trial. 'Switch' cues indicated that participants should switch to the other attentional state on the following trial. (**B**) The attention task involved the presentation of 3D-rendered rooms with paintings. Participants had to attend either to the style of the paintings ('art' trials) or the spatial layout of the rooms ('room' trials). On 'art' trials, the task was to find paintings that could have been painted by the same artist because of their similarity in artistic style, even though the content of the paintings might be different (e.g., the art match and base image have paintings by the same artist). On 'room' trials, the task was to find rooms that had the same spatial layout from a different perspective, even though their other features (wall color, specific furniture exemplars) varied (e.g., the room match and the base image have the same spatial layout from a different perspective). (**C**) Trial structure of the attention task. In the explicitly instructed task, the attentional state on each trial was randomly assigned ('ART' or 'ROOM'). On 'art' trials, participants had to determine if any of the paintings in the search set was painted by the same artist as the painting in the base image (i.e., if there was an art match). On 'room' trials, participants had to determine if any of the rooms in the search set had the same spatial layout as the room in the base image (i.e., if there was a room match). The memory-guided task was similar, except the attentional cue was not explicitly instructed at the beginning of each trial. Instead, participants had to choose their attentional goal at the beginning of each trial based on the stay or switch cue in the previous trial. Here, there is a room 'stay' cue (outlined in green), indicating that on the next trial, the participant should select 'room' as their attentional goal. If instead there was a room 'switch' cue, the participant would have to select 'art' as their attentional goal on the following trial. Particular stay and switch cues only appeared in the attended dimension: I.e., art stay/switch cues only appeared on trials where art was attended, and room stay/switch cues only appeared on trials where rooms were attended. Finally, some trials contained neither a stay cue nor a switch cue. On trials following such 'no cue' trials, participants were free to choose either 'art' or 'room' as their attentional state. Stay/switch cues were also embedded in the search set in the explicitly instructed task, but there they had no relevance for the upcoming attentional state.

The online version of this article includes the following figure supplement(s) for figure 1:

*Figure 1 continued on next page*

*Figure 1 continued*

**Figure supplement 1.** Analysis approaches for examining preparatory activity.

The *memory-guided* task had the same basic structure, except the attentional cue ('ART' or 'ROOM') was not overtly instructed at the beginning of each trial. Instead, attentional states were chosen by the participant based on stay and switch cues that were learned in an earlier phase of the experiment. Specifically, participants first learned four stimuli, two that signaled that they should stay in the same attentional state on the following trial ('stay cues') and two that signaled that they should switch to the other attentional state on the following trial ('switch cues'). During the subsequent attention task, a stay or switch cue could be embedded in the search set for any given trial. Thus, memory for the stay/switch cue on trial *N*, as well as memory for what that cue signaled, had to be used to guide attention on trial *N*+1.

In sum, we compared attention in two tasks: One where attentional goals were instructed at the beginning of each trial with an explicit cue, and one in which memory for specific images had to be used to select attentional goals. The tasks were identical otherwise — same stimuli, same motor demands — allowing us to rigorously test whether and how the hippocampus and vmPFC support memory-guided attention. Our main prediction was that these regions would prepare for upcoming attentional states, primarily when those states were guided by memory.

## Results

### Behavior

We first examined behavioral performance with two goals in mind: First, to determine if attention was effectively manipulated, and second, to determine if performance was roughly equivalent across the memory-guided and explicitly instructed tasks. This would ensure that differences in brain activity levels across the tasks are unlikely to be driven by differences in task difficulty (*Barch et al., 1997*; *McKiernan et al., 2003*).

To determine if attention was effectively engaged, we compared behavioral performance (A': 1 = perfect, 0.5 = chance, and response times) on valid vs. invalid trials. On valid trials, the attentional cue at the beginning of the trial — whether it was selected by the participant based on memory, or explicitly instructed — matched the probe at the end (e.g., participants were attending to room layouts, and at the end of the trial were probed as to whether there was a room match). On invalid trials, the attentional cue at the beginning of the trial did *not* match the probe at the end (e.g., participants were attending to room layouts, and at the end of the trial were probed as to whether there was an *art* match). If attention is effectively engaged by the cue at the beginning of the trial, participants should be more accurate and faster on valid vs. invalid trials. This should be the case whether the attentional cue was selected by the participant based on memory, or explicitly instructed.

We tested this with a 2-by-2 repeated measures ANOVA with the factors task (memory-guided, explicitly instructed) and cue validity (valid, invalid). Indeed, behavioral sensitivity (i.e., A' for detecting art or room matches) was higher on valid trials (M = 0.809, 95% CI [0.787, 0.831]) compared to invalid trials (M = 0.508, 95% CI [0.451, 0.565]), as revealed by a main effect of cue validity, $F(1, 28)$ =128.13, p<0.0001, $\eta_p^2$ = 0.82 (*Figure 2*). In fact, sensitivity was higher than chance only on valid trials (memory-guided: $t(28)$ = 20.25, p<0.0001, $d$ = 3.76, 95% CI [0.768, 0.828], explicitly instructed: $t(28)$ = 26.01, p<0.0001, $d$ = 4.83, 95% CI [0.795, 0.846]), and not on invalid trials (memory-guided: $t(28)$ = 0.66, p=0.51, $d$ = 0.12, 95% CI [0.412, 0.545], explicitly-instructed: $t(28)$ = 1.08, p=0.29, $d$ = 0.20, 95% CI [0.468, 0.606]). Moreover, response times were slower on invalid compared to valid trials, $F(1, 28)$=76.50, p<0.0001, $\eta_p^2$ = 0.73. These results suggest that our manipulation of attentional states was successful: Participants selectively attended to the category (art; room) that they chose in the memory-guided task and that they were instructed to attend in the explicitly instructed task.

We next examined behavioral performance across the memory-guided and explicitly instructed tasks, and found that the difference between them was not statistically significant (i.e., no main effect of task), $F(1, 28)$=3.20, p=0.084, $\eta_p^2$ = 0.10. The task by validity interaction was also not

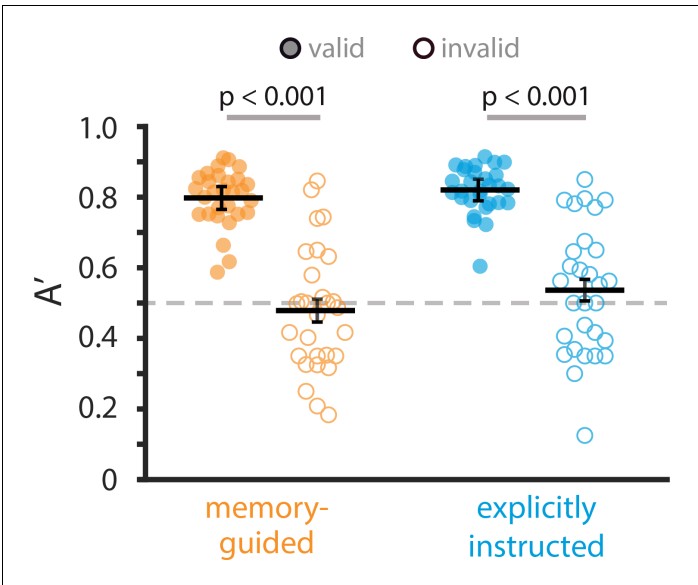

**Figure 2.** Behavioral results. Sensitivity (A') in detecting (art or room) matches, shown separately for each task (memory-guided, explicitly instructed) and for valid vs. invalid trials (filled and open circles, respectively). Circles are individual participants. Solid lines show average A' across participants, and error bars indicate the standard error of the mean for the within-participant valid – invalid difference. The dashed line indicates chance performance (A'=0.5). A' was higher on valid vs. invalid trials and was not significantly different between the memory-guided and explicitly instructed tasks.

significant, $F(1, 28)=1.11$, p=0.30, $\eta_p^2 = 0.04$. Because only valid trials were used in some fMRI analyses (see Methods), we also compared task performance on valid trials only. Again, the difference in A' for the memory-guided vs. explicitly instructed tasks was not statistically significant, $t(28) = 1.32$, p=0.20, $d = 0.25$, 95% CI [−0.058, 0.012]. Therefore, the tasks were of comparable difficulty, with similar modulations of attentional behavior by cue validity.

To ensure that, in the memory-guided task, individuals were indeed using the stay and switch cues to guide their attentional states, we examined their accuracy in choosing the correct attentional state based on the stay/switch cue in the previous trial (e.g., choosing 'room' as the attentional goal when the previous trial contained either a room 'stay' cue or an art 'switch' cue). Decision accuracy was high and was not significantly different between 'stay' cues (M = 0.949, 95% CI [0.932, 0.955]) and 'switch' cues (M = 0.967, 95% CI [0.954, 0.978]), $t(28) = 1.68$, p=0.10, $d = 0.31$, 95% CI [−0.004, 0.040]. Thus, participants were successfully able to use stay/switch cues to select memory-guided attentional goals.

## fMRI

### Activity enhancement for memory-guided vs. explicitly instructed attention

If the hippocampus and vmPFC are more involved in attentional behaviors that are guided by memory, then they should show enhanced univariate activity during the memory-guided vs. explicitly instructed task. To examine this, we compared BOLD activity in these regions during the attention task (i.e., when the images were on the screen and participants were attending to artistic style or room layout). Indeed, BOLD activity was higher for the memory-guided vs. explicitly instructed task in both hippocampus, $t(28) = 2.54$, p=0.017, $d = 0.47$, 95% CI [0.872, 8.125], and vmPFC, $t(28) = 3.74$, p=0.0008, $d = 0.69$, 95% CI [2.518, 8.611] (*Figure 3A*).

To determine if this difference in univariate activity is related to differences in behavioral performance across tasks, we examined whether A' differences on the memory-guided vs. explicitly instructed task predicted univariate activity differences between these two tasks, across individuals. This relationship was not statistically significant in hippocampus ($R^2 = 0.03$, p=0.39, 95% CI [−0.502, 0.214]) or in vmPFC ($R^2 = 0.02$, p=0.52, 95% CI [−0.470, 0.253]). Thus, univariate activity

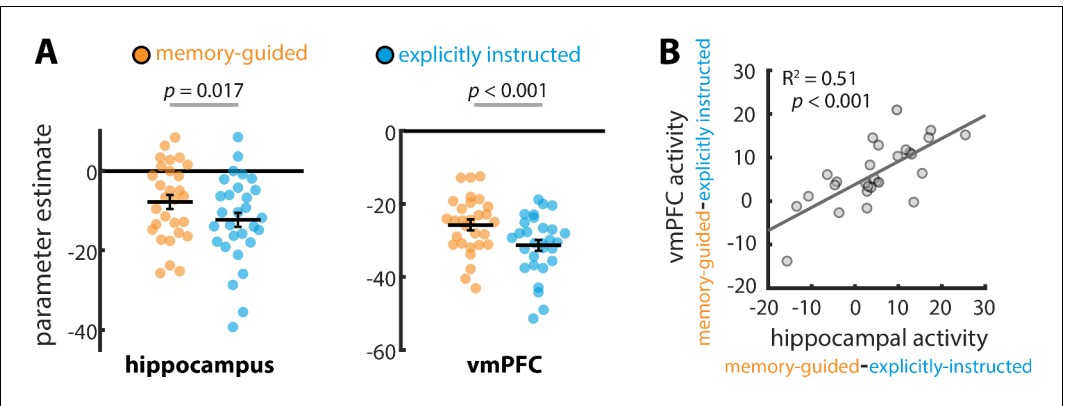

**Figure 3.** Univariate activity for memory-guided *vs.* explicitly instructed attention. (**A**) BOLD activity was higher for the memory-guided vs. explicitly instructed task, for both the hippocampus and vmPFC. Circles show parameter estimates (i.e., univariate BOLD activity) for individual participants. Solid lines show average parameter estimates across individuals, and error bars indicate standard error of the mean for the within-participant task difference (i.e., memory-guided – explicitly instructed). (**B**) The univariate activity enhancements for memory-guided attention (i.e., memory-guided parameter estimates – explicitly instructed parameter estimates) in the hippocampus and vmPFC were correlated across individuals.

enhancement in these regions for memory-guided attention cannot be explained solely by differences in behavioral performance.

If the hippocampus and vmPFC work together to establish memory-guided attentional states, then the extent to which one region's activity is modulated by memory-guided attention might predict how much the other region's activity shows such modulation. Indeed, the activity enhancement in each region for memory-guided attention (i.e., the BOLD activity difference for memory-guided vs. explicitly instructed tasks) was strongly correlated across individuals, $R^2$ = 0.51, p=0.000022, 95% CI [0.477, 0.867] (*Figure 3B*). Importantly, this correlation remained significant when controlling for individual differences in behavioral performance across tasks ($R^2$ = 0.50, p<0.0001). Together, these results suggest that the hippocampus and vmPFC play a similar functional role in memory-guided attention, and may be working together. In the Discussion, we further consider what enhanced univariate activity in these regions might reflect.

## Representations of current, and upcoming, attentional goals

Our primary question was whether the hippocampus and vmPFC can use memory to prepare for upcoming attentional states. Thus, we differentiate between two main periods on any given trial: (1) the *image period*, when images are on the screen and participants are actively attending to artistic style or room layout, and (2) the *orienting period*, when participants are pushing a button to initiate the trial and seeing the attentional cue (*Figure 1*). In the explicitly instructed task, participants simply choose which button to press, and then the attentional cue ('ART' or 'ROOM') is randomly assigned. In the memory-guided task, participants select 'art' or 'room' as the attentional state based on memory for the preceding trial. Based on prior studies showing that hippocampal univariate activity is enhanced in preparation for upcoming, memory-guided attentional goals (*Stokes et al., 2012*), we predicted that the hippocampus — and vmPFC, given their tight connection for memory-guided behavior (*Euston et al., 2012*; *Kaplan et al., 2017*; *Shin and Jadhav, 2016*) — would show preparatory coding during the orienting period. Specifically, we predicted that during the *orienting period*, activity patterns in the hippocampus and vmPFC would resemble the attentional state (i.e., art vs. room) that is upcoming in the *image period*, primarily when that attentional state was selected on the basis of memory.

In order to test this prediction, we first needed to establish that the hippocampus and vmPFC differentiate between the two attentional states (art vs. room) during the image period. This would then allow us to determine whether neural signatures of the art vs. room states appear in a preparatory fashion during the orienting period, particularly for memory-guided attention. Our past fMRI studies indicate that the hippocampus does indeed differentiate between the art vs. room states

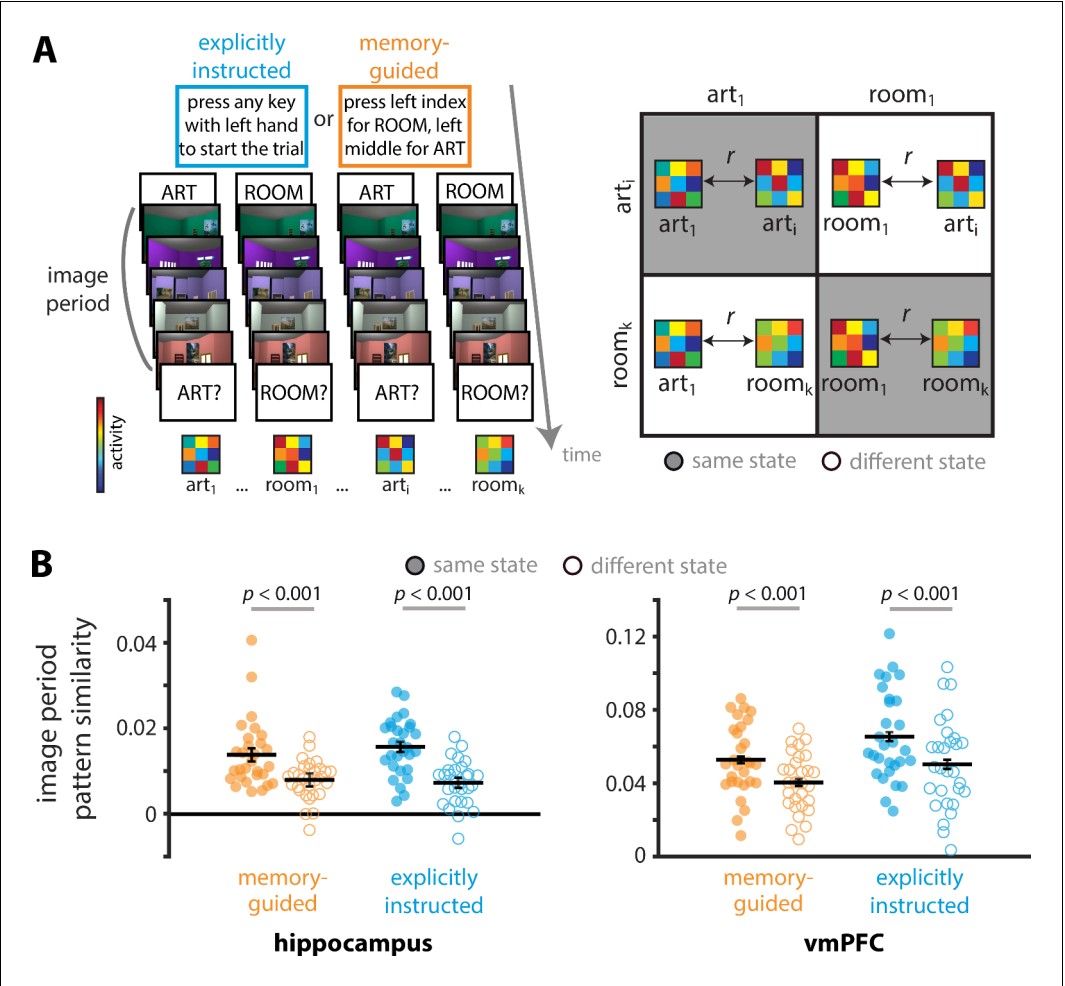

**Figure 4.** Representations of current attentional states. (**A**) Image period pattern similarity was calculated by correlating activity patterns across trials of the same vs. different attentional states, separately for each task. Here $art_1$, $room_1$, $art_i$, and $room_k$ indicate 1st art trial, 1st room trial, $i^{th}$ art trial, and $k^{th}$ room trial within a given task (memory-guided, explicitly instructed) respectively. Correlations were compared for trials of the same attentional state (i.e., art-art and room-room; right panel, gray background) and trials of different attentional states (i.e., art-room; right panel, white background). (**B**) Both the hippocampus and vmPFC represented current attentional states, with higher pattern similarity for trials of the same vs. different attentional states. Full circles and empty circles show pattern similarity for each participant for trials of the same state and different state, respectively. Solid lines show average pattern similarity across individuals. The results are shown as Pearson correlations, but statistical tests were performed after applying the Fisher transformation. The error bars indicate standard error of the mean for the within-participant attentional state difference (i.e., same - different) for each task.

during the image period (*Aly and Turk-Browne, 2016a*; *Aly and Turk-Browne, 2016b*), but here we sought to replicate this and extend it to vmPFC.

To this end, we obtained patterns of activity in the hippocampus and vmPFC for each image period, and then correlated these activity patterns as a function of the participants' attentional state (*Figure 4A*). We compared pattern similarity for trials of the same attentional state (i.e., art-art, room-room) to pattern similarity for trials of different attentional states (i.e., art-room) separately for the memory-guided and explicitly instructed tasks. If a brain region represents online attentional states, then pattern similarity should be higher for trials of the same state vs. trials of different states (*Aly and Turk-Browne, 2016a*; *Aly and Turk-Browne, 2016b*). This was the case for both the hippocampus and vmPFC, in both the memory-guided (hippocampus: $t(28) = 3.82$, p=0.00067, $d = 0.71$, 95% CI [0.003, 0.009], vmPFC: $t(28) = 6.58$, p<0.0001, $d = 1.22$, 95% CI [0.009, 0.017]) and explicitly instructed tasks (hippocampus: $t(28) = 7.12$, p<0.0001, $d = 1.32$, 95% CI [0.006, 0.011], vmPFC: $t(28)$

= 6.07, p<0.0001, *d* = 1.13, 95% CI [0.010, 0.021]). These results confirm that the hippocampus and vmPFC represent online attentional states (*Figure 4B*), a necessary precursor for examining preparatory attentional states during the orienting period.

Having confirmed that the hippocampus and vmPFC represent online attentional states (i.e., attentional states during the image period), we next tested whether these regions also represent *preparatory* attentional states — i.e., whether their activity patterns during the *orienting period* code for attentional states that are upcoming during the *image period*.

To that end, we first calculated 'template' patterns of activity by averaging activity patterns during the *image period* across trials, separately for the art and room attentional states (*Figure 5A*). These 'template' activity patterns indicate, for a given brain region, what the BOLD activity pattern looks like when participants are actively attending to artistic style vs. room layout in the 3D-rendered images. We then correlated activity patterns during each individual *orienting period* with the two templates, and binned these correlations based on whether the template matched the orienting period attentional cue (e.g., correlation between the art template and the orienting period activity pattern on an art trial) or mismatched (e.g., correlation between the room template and the orienting period activity pattern on an art trial). This was repeated for each trial, and the resulting correlations were averaged separately for the memory-guided and explicitly instructed tasks. Lastly, in order to obtain a measure of preparatory attentional state representations, we calculated the difference between match-to-same-template pattern similarity and match-to-different-template pattern similarity. If a brain region shows preparatory coding, its orienting period activity patterns should resemble the same-state template more than the different-state template.

Indeed, for the memory-guided task in the hippocampus (*Figure 5B*), activity patterns during the orienting period resembled the upcoming attentional state more than the other attentional state, *t*(28) = 4.78, p=0.00005, *d* = 0.89, 95% CI [0.008, 0.021]. Unexpectedly, this effect was also observed when attention was explicitly instructed, *t*(28) = 2.71 p=0.011, *d* = 0.50, 95% CI [0.001, 0.007]. Critically, however, preparatory attentional states in the hippocampus were stronger for the memory-guided vs. explicitly instructed task, *t*(28) = 3.18, p=0.004, *d* = 0.59, 95% CI [0.004, 0.017].

In vmPFC, activity patterns during the orienting period resembled the upcoming attentional state more than the other attentional state for both the memory-guided, *t*(28) = 6.25, p<0.00001, *d* = 1.16, 95% CI [0.010, 0.019], and explicitly instructed tasks, *t*(28) = 4.12, p=0.00030, *d* = 0.77, 95% CI [0.007, 0.020]. Contrary to our hypothesis, this effect was not significantly different between the tasks, *t*(28) = 0.77, p=0.45, *d* = 0.14, 95% CI [−0.003, 0.006]. Thus, the hippocampus, but not vmPFC, preferentially represented upcoming memory-guided vs. explicitly instructed attentional states.

## Robustness of preparatory attentional states

For the preceding analysis, we used common image period templates for the memory-guided and explicitly instructed tasks: Art trials from both tasks were used to create an art template, and room trials from both tasks were used to create a room template (*Figure 1—figure supplement 1*). This was done because using separate templates for each task might artificially create differences in orienting period pattern similarity values even if the orienting period patterns do not differ across tasks (e.g., different numbers of correct vs incorrect trials across tasks may lead to different template activity patterns). Furthermore, as in the image period analysis and our previous work (*Aly and Turk-Browne, 2016a*; *Aly and Turk-Browne, 2016b*), we only used valid trials for the image period templates (this was to prevent neural activity related to invalid probes from contaminating image period activity patterns). Finally, we analyzed all orienting periods in each task, whether the previous trial contained a stay/switch cue or did not contain one of these cues (i.e., 'no-cue' trials). (Note that in the explicitly instructed task, stay/switch cues were embedded in the search set but had no relevance for the attentional state on the following trial). We included no-cue trials because, in the memory-guided task, the attentional state decisions following these trials still had to be guided by memory: In order to know that the attentional goal could be chosen freely, participants needed to remember that no stay or switch cue was presented on the previous trial.

However, one could argue for alternatives to each of these decisions (*Figure 1—figure supplement 1*). For example, because the memory-guided and explicitly instructed tasks had different demands during the image period, it can be argued that separate image period templates should

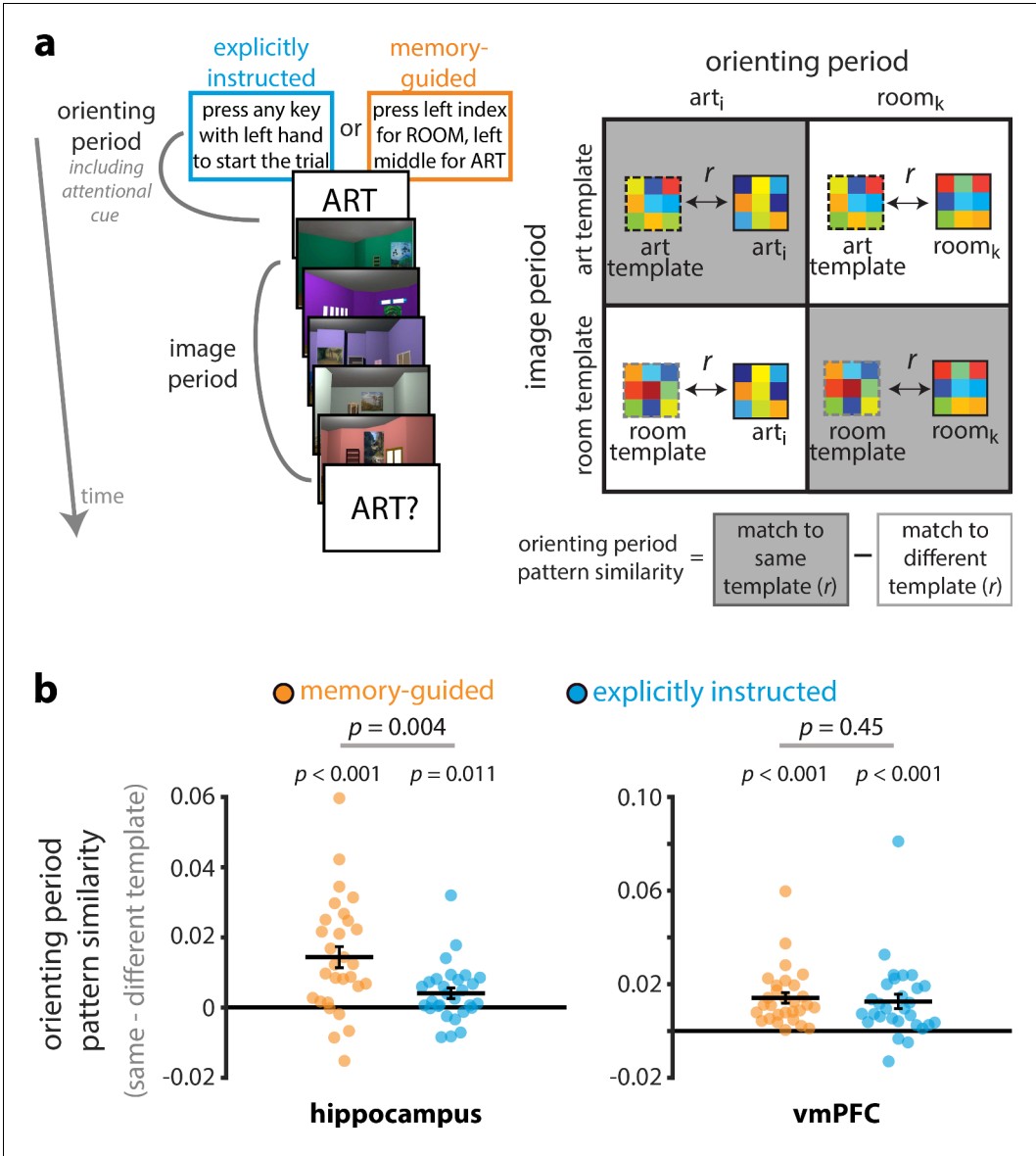

**Figure 5.** Representations of upcoming attentional states. (A) Art and room attentional state 'templates' were created by averaging image period activity patterns across trials, separately for art and room attentional states. Next, the orienting period activity pattern for each trial was correlated with these templates to obtain match to same template (e.g., room orienting period to room template) and match to different template (e.g., room orienting period to art template) pattern similarity values. Lastly, the match-to-different-template correlation was subtracted from the match-to-same-template correlation to obtain a measure of preparatory attentional state representations. (B) Pattern similarity values are shown as difference scores between the match-to-same-template correlation and the match-to-different-template correlation: More positive values indicate more evidence for the upcoming attentional state, and more negative values indicate more evidence for the other attentional state. Both the hippocampus and vmPFC showed preparatory coding, with orienting period activity patterns resembling the upcoming attentional state more than the other attentional state. In the hippocampus, this preparatory coding was stronger for memory-guided vs explicitly instructed attention. Circles and solid lines show individual and average pattern similarities, respectively. The results are shown as Pearson correlations, but statistical tests were performed after applying the Fisher transformation. The error bars indicate standard error of the mean for the within-participant difference in attentional state match (i.e., match to same template – match to different template) for each task.

The online version of this article includes the following figure supplement(s) for figure 5:

**Figure supplement 1.** Representations of upcoming attentional states following 'switch' cue trials.

*Figure 5 continued on next page*

*Figure 5 continued*

**Figure supplement 2.** Whole-brain searchlight analysis of preparatory attentional states during the orienting period.

be used for each task. Furthermore, contamination of image period brain activity by invalid probes should not differ across the memory-guided and explicitly instructed tasks, so one could argue for including invalid trials as well. Finally, attentional-state decisions following trials in which no stay or switch cue was presented might be less memory-driven than those following a stay or switch cue. This is because memory for the *particular* type of cue was required for choosing the correct attentional state following stay and switch cue trials, but memory for the mere *presence* or *absence* of a cue was sufficient following no-cue trials. Thus, one could argue that orienting periods following 'no cue' trials should be excluded from analyses. We therefore tested the robustness of our orienting period results by re-running the analyses with these alternative decisions.

We replicated the same pattern of results when: (i) using separate image period templates for the memory-guided vs. explicitly instructed tasks as opposed to a common template; (ii) using image period templates that include both valid and invalid trials as opposed to valid trials only, and (iii) analyzing only those orienting periods that followed either a stay cue or a switch cue (i.e., excluding orienting periods following no-cue trials).

Specifically, for the former analysis (i), we replicated the finding of preparatory attentional states in both hippocampus (memory-guided: $t(28) = 4.50$, p=0.00011, $d = 0.84$, 95% CI [0.007, 0.019]; explicitly-instructed: $t(28) = 2.26$, p=0.032, $d = 0.42$, 95% CI [0.0003, 0.006]), and vmPFC (memory-guided: $t(28) = 5.02$, p=0.00003, $d = 0.93$, 95% CI [0.008, 0.018]; explicitly-instructed: $t(28) = 3.79$, p=0.00073, $d = 0.70$, 95% CI [0.006, 0.019]). As in the main analysis, these preparatory attentional states were stronger for the memory-guided vs. explicitly instructed task in the hippocampus, $t(28) = 3.32$, p=0.0025, $d = 0.62$, 95% CI [0.004, 0.016], but did not significantly differ across tasks in vmPFC, $t(28) = 0.31$, p=0.76, $d = 0.06$, 95% CI [−0.005, 0.006].

For the second analysis (ii), we also replicated the finding of preparatory attentional states in both hippocampus, (memory-guided: $t(28) = 4.24$, p=0.00022, $d = 0.79$, 95% CI [0.007, 0.020]; explicitly-instructed: $t(28) = 2.69$, p=0.012, $d = 0.50$, 95% CI [0.001, 0.009]), and vmPFC (memory-guided: $t(28) = 6.29$, p<0.00001, $d = 1.17$, 95% CI [0.009, 0.018]; explicitly-instructed: $t(28) = 4.23$, p=0.00023, $d = 0.78$, 95% CI [0.006, 0.018]). Once again, preparatory attentional states were stronger for the memory-guided task in the hippocampus, $t(28) = 2.38$, p=0.024, $d = 0.44$, 95% CI [0.001, 0.015], and did not significantly differ between tasks in vmPFC, $t(28) = 0.74$, p=0.47, $d = 0.14$, 95% CI [−0.003, 0.006].

Finally, for the third analysis (iii), we again replicated the finding of preparatory attentional states for both the memory-guided, $t(28) = 6.56$, p<0.00001, $d = 1.22$, 95% CI [0.011, 0.021], and explicitly instructed tasks in vmPFC, $t(28) = 4.00$, p=0.00042, $d = 0.74$, 95% CI [0.007, 0.021]. In hippocampus, preparatory attentional states were again present for the memory-guided task, $t(28) = 4.14$, p=0.00029, $d = 0.77$, 95% CI [0.008, 0.025], but failed to reach significance in the explicitly instructed task ($t(28) = 1.84$, p=0.076, $d = 0.34$, 95% CI [−0.0004, 0.007]; note that this analysis is reduced in power because 1/3 of the trials were dropped). Once again, preparatory attentional states were stronger for the memory-guided task in the hippocampus, $t(28) = 3.16$, p=0.0038, $d = 0.59$, 95% CI [0.005, 0.022], but did not significantly differ across tasks in vmPFC, $t(28) = 0.70$, p=0.49, $d = 0.13$, 95% CI [−0.004, 0.007].

Thus, the main results are robust to many different analysis decisions. However, there is another potential concern. Are the observed results due to autocorrelation between orienting period and image period activity patterns, as a result of sluggish hemodynamic signals? We believe not, for several reasons. First, autocorrelation between the orienting period and image period should be higher for the explicitly instructed vs. memory-guided task because response times to initiate the trial were on average shorter for the explicitly instructed task (0.92 s vs. 1.12 s; $t(28) = 2.44$, p=0.021, $d = 0.45$, 95% CI [0.032, 0.361]). However, preparatory coding was stronger for the *memory-guided* task in the hippocampus and did not differ between tasks in vmPFC. Second, the image period templates — against which orienting period activity patterns were compared — were obtained from different runs of the task to remove within-run autocorrelation (*Mumford et al., 2014*). Third, the last brain

volume (TR) for the orienting period and the first brain volume for the image period were excluded from the analysis to reduce autocorrelation between the image period and orienting period signals. (Note that the last brain volume for the orienting period was not dropped if that was the only volume during which the attentional cue was presented). Thus, we argue that the preparatory attentional state representations observed in the orienting period are not simply the result of autocorrelation between orienting period and image period activity patterns.

One could argue that dropping the last brain volume for the orienting period activity pattern disadvantages the opportunity to detect preparatory coding for the explicitly instructed task more than the memory-guided task. This is because, for the memory-guided task, the attentional state for trial *N+1* is known as soon as trial *N* is over; but for the explicitly instructed task, it is only known when the attentional cue is presented at the end of the orienting period. To confirm that this is not the case, we re-ran the orienting period analysis including the last orienting period brain volume, and obtained the same pattern of results. We observed preparatory attentional states for both the memory-guided task (hippocampus: $t(28)$ = 4.84, p=0.00004, $d$ = 0.90, 95% CI [0.008, 0.021], vmPFC: $t(28)$ = 6.42, p<0.00001 $d$=1.19, 95% CI [0.010, 0.020]) and the explicitly instructed task (hippocampus: $t(28)$ = 3.11, p=0.0043, $d$ = 0.58, 95% CI [0.002, 0.008], vmPFC: $t(28)$ = 4.18, p=0.00026, $d$ = 0.78, 95% CI [0.007, 0.020]). Importantly, preparatory attentional states were stronger for the memory-guided vs. explicitly instructed tasks in the hippocampus, $t(28)$ = 3.04, p=0.00504, $d$ = 0.57, 95% CI [0.003, 0.017], and no difference in preparatory attentional states across tasks was measured in vmPFC, $t(28)$ = 0.86, p=0.40, $d$ = 0.16, 95% CI [−0.003, 0.006]. Together, these findings suggest that our results are robust and cannot be attributed to idiosyncratic analysis decisions.

## Retrieval of past states or preparation for upcoming states?

We argue that multivariate patterns of activity in the hippocampus during the orienting period reflect preparation for upcoming attentional states. However, is it possible that these activity patterns instead reflect retrieval of the attentional state from the previous trial? This is unlikely for the explicitly instructed task, where memory for the previous trial is not relevant for the attentional state on the current trial. Thus, preparatory signals for the explicitly instructed task in the hippocampus and vmPFC likely index anticipation of the upcoming task rather than memory retrieval. For the memory-guided task, however, it is possible that participants use the orienting period of a given trial to retrieve what they did on the previous trial. For example, during the orienting period for an upcoming 'room' trial, a participant may remember that the previous trial was a 'room' trial with a stay cue (or an 'art' trial with a switch cue). Are hippocampal activity patterns reflecting such memory retrieval?

Trials in which participants stay in the same attentional state as the previous trial are ambiguous: Remembering the previous trial and preparing for the current trial would be indistinguishable with our analysis because the attentional states are the same. However, trials in which participants switch from one attentional state to the other provide a strong test of our hypothesis. If hippocampal activity patterns during the orienting period reflect memory retrieval of the previous trial, they should resemble the previous attentional state more than the upcoming attentional state. If, however, hippocampal activity patterns during the orienting period reflect preparation, they should resemble the upcoming attentional state more than the previous one.

Indeed, when we analyzed only the trials that followed a switch cue (*Figure 5—figure supplement 1*), we found that hippocampal activity patterns during the orienting periods of the memory-guided task resembled the upcoming attentional state more than the other (previous trial's) attentional state, $t(28)$ = 2.90, p=0.0072, $d$ = 0.54, 95% CI [0.004, 0.023]. We also conducted this analysis for the explicitly instructed task for completeness (although, here, a switch cue has no relevance for the attentional state on the following trial). Here, we found no evidence for an attentional state representation (neither the upcoming attentional state nor the previous attentional state) during the orienting period, $t(28)$ = 0.16, p=0.87, $d$ = 0.03, 95% CI [−0.006, 0.007]. (We are cautious in over-interpreting this null effect because this analysis contains roughly one-third the trials in the main analysis, and hence has lower statistical power.) Finally, as in our main analysis, multivariate evidence for upcoming attentional states in the hippocampus was higher for the memory-guided vs. explicitly instructed task, $t(28)$ = 2.85, p=0.008, $d$ = 0.53, 95% CI [0.004, 0.023]. These results therefore suggest that, during the memory-guided task, hippocampal activity patterns during the orienting period

reflect preparation for the upcoming attentional state rather than retrieval of the preceding attentional state. This preparation for upcoming attentional states may involve memory retrieval of task-relevant goals and/or the use of these memories to bias neural processing toward task-relevant features. We discuss the content of such preparatory signals in more detail in the Discussion.

For completeness, we also analyzed only those trials following a switch cue for vmPFC and replicated our main results: Activity patterns during the orienting period resembled the upcoming attentional state more than the other (previous trial's) attentional state for both the memory-guided, $t(28)$ = 4.29, p=0.00019, $d$ = 0.80, 95% CI [0.009, 0.027] and explicitly instructed tasks, $t(28)$ = 4.45, p<0.0001, $d$ = 0.83, 95% CI [0.009, 0.023]. These preparatory states did not significantly differ across tasks, $t(28)$ = 0.71, p=0.48, $d$ = 0.13, 95% CI [−0.004, 0.008]. These results suggest that for vmPFC — as for hippocampus — activity patterns during the orienting period reflect preparation for the upcoming attentional state, rather than retrieval of the previous attentional state.

## Hippocampal interactions with visual cortex

Our results so far indicate that the hippocampus is more strongly engaged by memory-guided vs. explicitly instructed attention (*Figure 3*) and represents both current (*Figure 4*) and upcoming (*Figure 5*) attentional states. Moreover, the hippocampus shows stronger preparation for memory-guided attention. How does the hippocampus transform memory cues in the environment (i.e., stay/switch cues) into preparatory attentional signals? One possibility is that hippocampal interactions with visual cortex are enhanced when memory must be used to guide attention. This would allow mnemonically relevant information in the environment to be detected via hippocampal-visual cortex communication. Once this information is detected, the hippocampus can then use it to prepare for attentional states that are guided by those mnemonic cues. To test this, we examined whether functional coupling between the hippocampus and visual cortex is enhanced for memory-guided attention. Because detection of stay/switch cues requires being in a task-relevant attentional state, we hypothesized that the attentional states of the hippocampus and visual

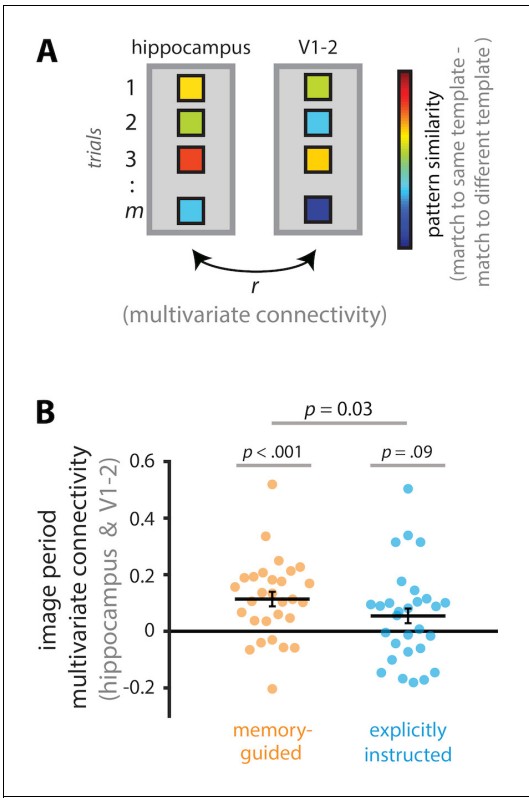

**Figure 6.** Multivariate connectivity between the hippocampus and V1-2. (**A**) To calculate multivariate connectivity, we first created art and room attentional state 'templates' by averaging image period activity patterns across trials, separately for art and room attentional states. Second, these templates were correlated with activity patterns for individual trials, separately for same (i.e., art trial-art template, room trial-room template) and different (i.e., art trial-room template, room trial-art template) attentional states. Third, for each trial, we calculated a measure of multivariate attentional state 'quality' by subtracting its correlation with the different state template (e.g., art trial-room template) from its correlation with the same state template (e.g., art trial-art template). These steps were performed separately for the hippocampus and V1-2. Lastly, we computed multivariate connectivity between the hippocampus and V1-2 by correlating their multivariate attentional state 'quality' scores across all trials. (**B**) Multivariate connectivity was greater than zero for the memory-guided task, but not different from zero for the explicitly instructed task, and the difference between tasks was statistically significant. The results are shown as Pearson correlations, but statistical tests were performed after applying the Fisher transformation. Circles and solid lines show individual-participant and average multivariate connectivity values, respectively. The error bars indicate standard error of the mean for the within-participant task difference (i.e., memory-guided - explicitly instructed).

cortex will be more strongly aligned for the memory-guided task.

To examine this, we capitalized on novel neuroimaging methods that allow investigation of multivariate coupling between regions: *multivariate* (or *informational*) *connectivity* (*Aly and Turk-Browne, 2016b*; *Anzellotti and Coutanche, 2018*; *Coutanche and Thompson-Schill, 2013*). We focused on visual areas V1-2 because representations in these regions are correlated with those in the hippocampus during memory retrieval and predictive coding (*Bosch et al., 2014*; *Hindy et al., 2016*).

We first determined the quality of attentional states in the hippocampus and V1-2 on a trial-by-trial basis. This was achieved by determining whether activity patterns on a given trial more strongly aligned with the task-relevant vs. task-irrelevant attentional state (e.g., on a trial with a 'good' room attentional state, hippocampal activity patterns should more strongly resemble the average room-state activity pattern vs. the average art-state activity pattern). We then correlated these measures of attentional state 'quality' across the hippocampus and V1-2.

Prior to measuring multivariate connectivity, we first had to confirm that V1-2 represents current attentional goals (a precursor to examining the covariation of attentional states between regions is that each region must represent attentional states; see *Figure 4*). Indeed, in V1-2, pattern similarity was higher for trials of the same attentional state vs. trials of different attentional states, for both memory-guided, $t(28) = 9.32$, p<0.0001, $d = 1.73$, 95% CI [0.092, 0.144], and explicitly instructed tasks, $t(28) = 11.83$, p<0.0001, $d = 2.20$, 95% CI [0.103, 0.146]. Next, we computed multivariate connectivity between the hippocampus and V1-2, as described above (*Figure 6A*). High multivariate connectivity (i.e., inter-regional correlation) indicates that when one region is in a 'good' attentional state, the other region is also in a good attentional state, and when one region is in a 'bad' attentional state, the other region is also in a bad attentional state. Multivariate connectivity was significantly above zero in the memory-guided task, $t(28) = 4.28$, p=0.00020, $d = 0.80$, 95% CI [0.061, 0.173], but not in the explicitly instructed task, $t(28) = 1.78$, p=0.086, $d = 0.33$, 95% CI [−0.009, 0.123]. The difference between tasks was statistically significant, $t(28) = 2.28$, p=0.030, $d = 0.42$, 95% CI [0.006, 0.114]. These findings raise the possibility that covariation in attentional states between the hippocampus and early visual cortex may enable mnemonically relevant information in the environment to be detected, and then acted upon, to guide behavior on the basis of memory (*Figure 6B*).

## Other measures of neural preparation

We have focused on multivariate measures of preparatory coding in the hippocampus: The extent to which orienting period activity patterns contain information about upcoming attentional states. Yet, a previous study found univariate activity enhancements in the hippocampus when memory was used to prepare for upcoming attentional goals (*Stokes et al., 2012*). In that study, hippocampal activity was enhanced when information in memory was available about an upcoming target location, even prior to the onset of attentional search. Here, we found that hippocampal activity levels are enhanced for memory-guided vs. explicitly instructed attention during the *image period* (*Figure 3*), but to more closely parallel the *Stokes et al. (2012)* study, we also examined whether univariate activity is enhanced during the *orienting period*, i.e., in anticipation of the attentional search task. However, during the orienting period, univariate activity in the hippocampus was not significantly different for memory-guided (M = 14.144, 95% CI [9.123, 19.164]) vs. explicitly instructed attention (M = 17.740, 95% CI [11.077, 24.402]), $t(28) = 1.28$, p=0.21, $d = 0.24$, 95% CI [−9.369, 2.176]. For completeness, we also examined univariate activity in vmPFC during the orienting period, but again found no significant difference between the memory-guided (M = 12.044, 95% CI [6.219, 17.870]) and explicitly instructed tasks, (M = 14.563, 95% CI [8.447, 20.678]), $t(28) = 0.85$, p=0.40, $d = 0.16$, 95% CI [−8.593, 3.556]. We return to this difference between the results of our study and those of *Stokes et al. (2012)* in the Discussion.

## Attentional preparation in other brain regions

Although our focus has been on the hippocampus and vmPFC, we conducted exploratory whole-brain analyses to investigate neural signatures of attentional preparation elsewhere in the brain. We used a searchlight approach to find brain regions whose orienting period activity patterns were significantly correlated with their image period activity patterns. This approach was used to look for regions that showed greater preparation for memory-guided vs explicitly instructed attention, and

regions that showed preparatory coding for either task treated separately. No voxels survived correction for multiple comparisons (p<0.05 family-wise error corrected) when looking for regions that showed greater preparation for memory-guided vs explicitly instructed attention. When we looked for preparatory coding for each task separately, a few isolated voxels survived correction for multiple comparisons but no meaningful clusters emerged (*Figure 5—figure supplement 2*). These results must of course be treated with caution: it is very likely that brain areas other than the hippocampus and vmPFC prepare for upcoming attentional goals, but more targeted region-of-interest analyses are required to uncover them.

## Discussion

### Summary

In daily life, we often use our memories to guide attention. For example, we use memory to decide where to attend when we navigate familiar routes, or which parts of the street to avoid because of dangerous potholes. However, attention in laboratory studies is typically investigated by providing explicit instructions to participants about what or where to attend. To bridge real-world behavior and laboratory studies, we explored the neural mechanisms underlying memory-guided vs. explicitly instructed attention. We designed two tasks that differed only in their requirement to use memory to guide attention. In the *explicitly-instructed* attention task, participants were given randomly determined attentional goals on each trial. In the *memory-guided* attention task, participants chose their attentional goals based on cues that had to be stored in memory. Based on previous studies implicating the hippocampus and vmPFC in memory-guided behaviors (*Euston et al., 2012*; *Kaplan et al., 2017*; *Shin and Jadhav, 2016*), we predicted that these regions would support the ability to use memory to prepare for anticipated attentional states.

Extending prior work (*Stokes et al., 2012*; *Summerfield et al., 2006*), we found that activity levels in both hippocampus and vmPFC were higher for memory-guided vs. explicitly instructed attention. Furthermore, the memory-guided activity enhancements in hippocampus and vmPFC were correlated across individuals, suggesting that these regions may play a common role or work together for memory-guided attention.

To further examine their role in memory-guided attention, we used representational similarity analyses (*Kriegeskorte et al., 2008*) to identify the information present in these regions in preparation for, and during, attentional guidance. Activity patterns in the hippocampus and vmPFC contained information about current and upcoming attentional states. Importantly, in the hippocampus, preparatory attentional state representations were stronger for memory-guided vs. explicitly instructed attention. Further analyses confirmed that these preparatory attentional states did not reflect retrieval of past attentional goals, but rather the anticipation of upcoming attentional states. Lastly, the hippocampus and early visual cortex (V1-2) showed increased covariation in their attentional state representations in the memory-guided vs. explicitly instructed task.

Together, these results elucidate how the hippocampus and vmPFC support memory-guided attention, and show that the hippocampus is preferentially involved in preparing for anticipated attentional goals that are guided by memory. Its role in memory-guided attention may be supported via its interactions with early visual cortex. These interactions may be the means by which mnemonically relevant information in the environment is detected and used to guide attention and perception. Thus, our work demonstrates the adaptive function of memories by highlighting the mechanisms by which past experiences can be used to prepare for future behaviors (*Nobre and Stokes, 2019*).

### Relation to prior studies

Many studies of memory have focused on the importance of the hippocampus and vmPFC for memory-guided behaviors, such as navigational decisions (*Euston et al., 2012*; *Kaplan et al., 2017*; *Shin and Jadhav, 2016*). Because the world is complex and contains many more features than those that are currently relevant for our needs, memory can only guide effective behavior insofar as it can guide attention. Yet, studies of attention almost entirely ignore memory systems of the brain, and instead focus on sensory regions and frontoparietal control networks (e.g., *Corbetta et al., 2005*; *Ester et al., 2016*; *Serences et al., 2005*). To determine how memories can flexibly guide behavior,

we must understand how memories, and memory systems of the brain, guide attention. We suggest that representations in, and coordination between, the hippocampus, early visual cortex, and vmPFC allow past experiences to trigger anticipation of upcoming attentional targets. In this way, memories of the past can be used to prepare for, and behave adaptively in, predicted environments.

Our work therefore complements prior studies on predictive coding in the hippocampus (*Hindy et al., 2016*; *Kok et al., 2012*). Many such studies, however, focus on the representation of future navigational trajectories or navigational goals (*Brown et al., 2016*; *Johnson et al., 2007*; *Pfeiffer and Foster, 2013*). Here, we show that non-navigational, abstract attentional states are also represented in the hippocampus in a preparatory manner. To our knowledge, our study is the first to show that the hippocampus and vmPFC can prepare for anticipated attentional states. In this way, the current work takes principles and findings from research on memory and discovers their applicability to goal-directed attention.

The current study also broadens the research literature on hippocampal contributions to attention (*Aly and Turk-Browne, 2017*). We have previously shown that attention modulates hippocampal representations (*Córdova et al., 2019*) and that this modulation predicts both online attentional behavior (*Aly and Turk-Browne, 2016a*) and memory formation (*Aly and Turk-Browne, 2016b*). Furthermore, hippocampal damage impairs performance on attention tasks that require processing of spatial relations (*Ruiz et al., 2020*). However, these studies are limited because they investigate attentional behaviors that are explicitly instructed, and thus are less ecologically valid than studies of memory-guided attention. Here, we expand on the contributions of the hippocampus to attentional behaviors by investigating scenarios in which attentional goals must be decided on the basis of past experience.

Our work was inspired by studies of memory-guided attention (e.g., *Stokes et al., 2012*; *Summerfield et al., 2006*) but it differs from them in a number of ways. One key difference is that many of these prior studies involved teaching participants the relationship between particular memory cues (e.g., scenes) and locations to be attended. Thus, participants were able to use memory to guide spatial attention, with knowledge of what visual content will be experienced. In contrast, participants in our study learned that particular memory cues signaled to either stay in the same attentional task or switch to a different one. This is akin to studies in which learned attention cues direct individuals to either hold or shift their current attentional focus (e.g., *Chiu and Yantis, 2009*; *Greenberg et al., 2010*; *Yantis et al., 2002*). Furthermore, the current study involved some trials in which participants were free to choose what to attend; this is similar to studies investigating the neural correlates of self-directed attentional decisions (*Taylor et al., 2008*). Although our study shares similarities with these latter investigations, it differs from studies of memory-guided attention in that memory did not allow individuals to anticipate specific visual content. Instead, it enabled participants to anticipate the upcoming task and, at a high-level, the types of visual features relevant for that task.

Despite these differences, however, prior studies and ours share similarities. First, like other studies of attention, we found that manipulations of attentional cue validity led to robust behavioral consequences (*Posner, 1980*; *Stokes et al., 2012*; *Summerfield et al., 2006*): participants were faster and more accurate on valid vs. invalid trials, and their performance on invalid trials was not different from chance. Thus, although our study manipulates a more abstract form of attention relative to other studies, it replicates a key behavioral marker that is used as evidence for an attentional manipulation. Second, our study converges with other studies of memory-guided attention in suggesting that the hippocampus plays a role in guiding attentional behaviors on the basis of past experience (see *Aly and Turk-Browne, 2017*, for a review).

For example, during the attentional search task (i.e., during the image period), hippocampus and vmPFC univariate activity levels were higher for memory-guided vs. explicitly instructed attention (*Figure 3*). This finding broadly replicates other studies of memory-guided attention, but enhanced univariate activity is somewhat ambiguous. Here, this difference could be a result of the demand to monitor the search set for remembered stay/switch cues, identify the meaning of those stay/switch cues, or it could reflect another cognitive process arising from the dual-task nature of the memory-guided condition. Thus, many potential cognitive functions can account for the univariate activity enhancement in hippocampus and vmPFC during memory-guided attention in this study.

We also found that these regions showed no difference in univariate activity levels between the memory-guided and explicitly instructed conditions during the orienting period. This null univariate

effect is in contrast to previous studies of memory-guided attention, which observed higher univariate activity in the hippocampus during preparation for memory-guided attention (*Stokes et al., 2012*). Why might there be this difference between our findings and those of *Stokes et al. (2012)*? One potential reason is the difference in information provided by memory. In *Stokes et al. (2012)*, the memory cues carried content-related information about target items: the cues signaled where in space a target will appear. Conversely, the memory (stay/switch) cues in the current study (indirectly) signaled the task that will be carried out on the upcoming trial, with no indication of specific visual content or targets that would appear. Furthermore, there was a long and variable blank delay between the orienting period and the attentional task in the *Stokes et al. (2012)* study; in the current study, the length of the orienting period was variable, but there was no blank delay between it and the attentional task. Thus, differences in the kind of information carried by memory (specific content vs. abstract task set), as well as in the timing of the orienting periods and the attention task, could have led to the observed differences in univariate activity during preparatory attention.

That said, another difference could be in the relative timing of memory retrieval in the two tasks. In order to use memory to anticipate upcoming attentional goals, one must first retrieve the relevant memory and then use it to prepare for the upcoming task at hand. The retrieval of an attentional goal and the use of this goal to prepare for upcoming tasks may be inextricably intertwined, but they may also be partly dissociable in time. One possibility, although speculative, is that hippocampal activity enhancements reflect memory retrieval of particular associations (as in *Stokes et al., 2012*), and such memory retrieval occurred earlier in our task vs. that of *Stokes et al. (2012)*. Specifically, it is possible that individuals retrieved the meaning of stay/switch cues before the orienting period, e.g., during the inter-trial interval or during the previous trial. This retrieved information may then be used to prepare for upcoming attentional states during the orienting period. Indeed, the image-period univariate activity enhancement in the hippocampus for memory-guided attention may reflect such memory retrieval (*Figure 3*). Future studies using methods with high temporal resolution (e.g., MEG/EEG) will be useful for determining the temporal dynamics by which the hippocampus switches from retrieving a past memory to using that memory to anticipate upcoming attentional states — if indeed, these are separable processes as opposed to inherently linked.

One final possibility for the different findings in our study and that of *Stokes et al. (2012)* is that univariate activity and multivariate activity patterns in the hippocampus are differentially sensitive to different kinds of information, e.g., retrieval of specific memories (*Stokes et al., 2012*) vs. abstract task sets (current study). Although once again speculative, this could potentially help explain why we observed effects during the orienting period in multivariate activity patterns but not overall univariate activity. Such a dissociation in the information present in univariate activity vs. pattern similarity is consistent with the finding that multivariate attentional state representations are dissociable from changes in overall activity levels (*Aly and Turk-Browne, 2016a*).

## Nature of preparatory attentional states

When a brain region prepares for, or anticipates, an upcoming task, what is being represented? We have referred to the orienting period activity patterns in hippocampus and vmPFC as reflecting preparatory attentional states. This is because activity patterns prior to, or in preparation for, an upcoming attentional task resembled those during the task itself. However, a number of different cognitive processes can lead to overlap in brain representations for engaging in a task and anticipating it. We consider these below.

One possibility is that preparatory attentional states observed in our study reflect the anticipated difficulty of art and room attentional states. For example, if a participant finds attending to art more challenging than attending to rooms, they may modulate arousal or effort when anticipating an art trial. This modulation of arousal or effort may have an effect on activity patterns in the hippocampus or vmPFC. As a result, activity patterns during the anticipation and execution of an art trial would be similar due to shared effort- or arousal-related components. If this is the case, individuals who found one attentional state much more difficult than the other (e.g., art harder than room or vice versa) should show stronger evidence of neural preparation. However, we did not find any significant correlations between performance differences on art and room trials and the strength of anticipatory attentional state representations (all *p*s > 23). Thus, we argue that differences in difficulty between art and room trials are unlikely to be the driving factor for pattern similarity across the orienting period and image period. That said, differences in *subjective* assessments of difficulty may

nevertheless contribute to the extent of neural preparation, even if *objective* performance differences do not seem to.

Previous studies have shown preparatory coding for concrete shapes and locations in the hippocampus and sensory regions (*Battistoni et al., 2017*; *Corbetta et al., 2005*; *Hindy et al., 2016*; *Kok et al., 2012*; *Stokes et al., 2009*). Preparatory representations of anticipated shapes or locations, in turn, are thought to facilitate the perception of task-relevant information in the external world (*Battistoni et al., 2017*). Is the preparatory coding observed in our study indicative of the brain's anticipation of particular objects or locations, or is it more abstract in nature?

Accordingly, another possibility is that participants, upon anticipating an art or room attentional state, start to represent concrete visual features related to those categories. For example, they might bring to mind paintings or rooms that were previously seen in the experiment. However, this approach may not be effective, because the particular paintings or rooms imagined are unlikely to be the specific ones relevant on that trial (because of the large number of images used in the experiment). A mismatch between imagined visual features and those that end up being relevant might hurt performance instead of boosting it. As a result, it may not be adaptive for individuals to bring to mind specific paintings or rooms in preparation for the upcoming attentional state. Instead, it may be beneficial to prioritize the visual system and hippocampus to process spatial/global information in general (for the room task) or color/object/local information in general (for the art task).

Thus, the preparatory attentional states that we observed may be relatively abstract in nature. This is particularly likely because the presence of these preparatory states was established by examining the similarity between activity patterns related to preparation (during the orienting period) and activity patterns related to attentional guidance (during the image period). Given that these image period activity patterns were calculated across trials that used many different visual images, they presumably reflect attentional states that are abstracted away from specific visual features on any given trial. However, what those abstractions are is not clear from the current study. The preparatory signals in hippocampus and vmPFC might reflect an abstract attentional orientation (attend to local features vs. global features; attend to color vs. geometry), maintenance of a task instruction (find a similar painting vs. find a similar room), or even a metacognitive state ('The art task is harder for me, so I should expend more effort'). As long as these cognitive processes occur during both the orienting period and the image period, they may be components of the observed preparatory signals. The representational nature of the preparatory attentional states that are observed in the present study therefore deserves further investigation.

One key limitation of the current study is the absence of a long period of no visual stimulation between the orienting period and the image period. A long blank period would have allowed cleaner isolation of preparatory signals from those related to carrying out the task itself. However, several measures were taken to reduce autocorrelation when comparing activity patterns from the orienting period to those from the image period, and we argue that the current results are difficult to explain with autocorrelation (see *Robustness of preparatory attentional states* and *Methods*). Nevertheless, it would be ideal for future studies to include a longer delay between the orienting period and image period, for better isolation of anticipatory neural states. This would be particularly useful if fMRI were complemented with EEG, to incorporate the high temporal resolution of the latter method (e.g., *Stokes et al., 2012*).

## What kind of memory is used to guide attention?

Attention can be guided by many forms of memory at multiple timescales (*Nobre and Stokes, 2019*). Which are at play in the current study? We believe that long-term memory, intermediate-term memory, and working memory all contribute. We elaborate on these below.

Long-term memory plays an essential role in our memory-guided task because the stay/switch cues that were used to select attentional states were well-learned ~30 min prior to the fMRI scan. Participants showed near-perfect performance in using these cues to select the correct attentional state. Moreover, the ability to detect art or room matches did not differ between the memory-guided and explicitly instructed tasks (*Figure 2*), suggesting that the additional demand to identify stay/switch cues in the memory-guided task might have been relatively automatized (*Logan, 1988*). Therefore, the long-term memories used to identify the stay/switch cues and retrieve their meanings were well-learned, and possibly partly semanticized. Indeed, semantic memories can contribute to the guidance of attention (*Brockmole and Le-Hoa Võ, 2010*; *Moores et al., 2003*; *Olivers, 2011*;

*Torralba et al., 2006*). This is common in daily life, where many cues that are used to direct attention (e.g., traffic signs) are extensively practiced and retained in semantic memory. However, memories for the stay/switch cues in the current study are likely not semantic to the same extent as memories for traffic signs, the latter of which are learned and practiced over a lifetime rather than ~30 min. Thus, although the stay and switch cues were well-learned, they were learned the same day as the fMRI scan and thus unlikely to be truly semanticized. Instead, they might more closely resemble episodic memories.

The second timescale of memory that may have contributed to attentional guidance in the current study lies somewhere between long-term and working memory: the relatively intermediate-term memory for what occurred on the previous trial. Specifically, when a new trial starts, participants have to remember their attentional state on the previous trial, and whether there was a stay or switch cue in the previous trial, to select their attentional state. Alternatively, participants may decide their attentional state for the following trial as soon as they see a stay/switch cue, and then store the intention in memory until the following trial starts. This memory — whether it is a memory for the intention or a memory for the stay/switch cue — might be stored as an episodic trace during the inter-trial interval and recalled at the beginning of the next trial. This would be consistent with work demonstrating that episodic memories can bias attention (*Stokes et al., 2012*; *Summerfield et al., 2006*). Alternatively, this information may be maintained in working memory throughout the inter-trial interval until the onset of the following trial.

Finally, once an individual decides what to attend to — or is told what they should attend to based on an explicit instruction — this attentional state is likely represented in working memory over the course of visual search. Indeed, attentional templates stored in working memory guide attention and bias perception in a way that aligns with attentional goals (*Carlisle et al., 2011*; *Chelazzi et al., 1998*; *Desimone, 1996*; *Gunseli et al., 2014a*; *Gunseli et al., 2014b*; *Olivers et al., 2011*; *Gunseli et al., 2016*). This form of working-memory-guided attention should contribute to performance in both the memory-guided and explicitly instructed tasks.

In sum, multiple timescales of memory likely contributed to performance in the current task (*Hutchinson and Turk-Browne, 2012*; *Nobre and Stokes, 2019*): long-term, overlearned memories; intermediate-term episodic memories; and working memory. Future studies will be useful for understanding the similarities and differences between attentional guidance by memories at these timescales. For example, one question is whether the hippocampus can be involved in the guidance of attention by semantic memories (e.g., when detecting and responding to a traffic sign) or if it is preferentially involved when episodic memories guide attention (e.g., when avoiding a pothole that we noticed yesterday). Such a question can also help better isolate the complementary roles of the hippocampus and vmPFC in memory-guided attention. It is possible that more semanticized or consolidated episodic memories might call on vmPFC to guide attention, while the hippocampus is more important for the guidance of attention by relatively recent or rich episodic memories. This would be consistent with the differential role of these regions in semanticized vs. vivid episodic memories (*Bonnici and Maguire, 2018*; *Sekeres et al., 2018*).

## Future directions

The current study confirmed our hypothesis that the hippocampus and vmPFC prepare for upcoming attentional states. However, contrary to our hypotheses, only the hippocampus — and not vmPFC — showed stronger preparation for memory-guided attention. Why might this be? There are at least two possible explanations. First, vmPFC might weight explicit instructions and memories equally when preparing for upcoming task goals, while the hippocampus may prioritize information that is retrieved from memory. Given the importance of the hippocampus for memory retrieval, it is reasonable that information that arises from within the hippocampus itself might, at least in some situations (*Tarder-Stoll et al., 2020*) be prioritized relative to information from the external environment.

An alternative possibility is that the hippocampus is capable of preparing for upcoming attentional states equally strongly regardless of how these states are guided (i.e., by memories vs. explicit instructions) — but we were not able to observe this in our task because of limitations of the experimental design. In particular, the upcoming attentional state was known for longer in the memory-guided vs. explicitly instructed task: attentional states for trial N were known as soon as trial N - 1 was over for the memory-guided task, but only known when the attentional cue was displayed on trial N for the explicitly instructed task. Furthermore, the attention task started relatively soon after

the attentional cue was shown. Thus, it is possible that vmPFC is able to rapidly prepare for upcoming attentional states regardless of how they are known, but the hippocampus needs more time in order to represent attentional goals that are cued by the environment. Future studies that use methods with higher temporal resolution (e.g., EEG/MEG), and longer delays between when attentional goals are known and when they must be used, will be needed to explore this question. Such methods can establish the temporal dynamics by which memory-guided vs. explicitly instructed attention influence representations across different brain regions.

What is the benefit of preparatory attentional states? Previous research has shown that representations in early visual cortex are sharpened for anticipated stimuli (e.g., *Kok et al., 2012*). Furthermore, attentional modulation of early visual cortex can bias the detection of goal-relevant information over distractors (*Peelen and Kastner, 2011*; *Reynolds et al., 1999*; *Stokes et al., 2009*). Such a biasing process has primarily been studied when attention is explicitly instructed. When attention is guided by memory, the hippocampus might be important for preparing visual cortex for task-relevant features (*Stokes et al., 2012*). For example, hippocampal anticipation of upcoming attentional states might enable visual cortex to prioritize the processing of task-relevant information. Indeed, hippocampal pattern completion is associated with predictive coding in early visual cortex (*Hindy et al., 2016*). The potential importance of hippocampal interactions with visual cortex for memory-guided attention was also evident in our study: The attentional states of hippocampus and early visual cortex were more strongly coupled for memory-guided vs. explicitly instructed attention. Such covariation may allow mnemonically relevant information to be detected in the environment, and then subsequently used by the hippocampus to prepare for upcoming attentional states. Future studies that investigate the direction of information flow between hippocampus and early visual cortex can test whether visual cortex first influences the hippocampus to cue the retrieval of relevant information, and whether this direction of influence reverses once hippocampal memories can be used to anticipate attentional states (*Place et al., 2016*).

We have largely considered the complementary functions of attention and memory: how memories can be used to guide attentional behavior. Yet, there can also be a tension between attention and memory, particularly when attention to the external world has to be balanced against the processing of internally retrieved memories. How does the hippocampus balance the demand between externally and internally oriented attention? This is particularly interesting to examine in cases like the current study, where both external attention and memory retrieval are needed for the effective guidance of behavior. One hypothesis is that the hippocampus might rapidly fluctuate between internal and external modes, prioritizing either attention/encoding or memory retrieval at different timepoints (*Hasselmo, 1995*; *Hasselmo and Fehlau, 2001*; *Hasselmo and Schnell, 1994*; *Hasselmo et al., 1996*; *Honey et al., 2018*; *Meeter et al., 2004*; *Patil and Duncan, 2018*; *Tarder-Stoll et al., 2020*). Although there are 'background' fluctuations between external and internal attention in the hippocampus, top-down goals or external factors (e.g., surprise) can also affect these fluctuations (*Sinclair and Barense, 2019*). Thus, one possibility is that the appearance of a stay/switch cue briefly switches the hippocampus from an externally oriented state to an internally focused one. Future studies will be needed to explore how the demands of internal and external attention are balanced by the hippocampus in the context of memory-guided attention.

## Conclusions

Memories frequently guide attention in the real world, but how they do so is relatively underexplored. We have shown that the hippocampus and vmPFC prepare for anticipated attentional states, and the hippocampus does so more strongly for attentional states that are selected on the basis of memory. Furthermore, attentional states in the hippocampus correlate, on a trial-by-trial basis, with those in early visual cortex when attention is guided by memories. This informational connectivity may be essential for enabling perceptual signals to cue memory-guided goals and for memory-guided goals to bias perception. Together, these findings suggest that memories can be flexibly used to guide attentional behavior, and that this process calls on representations in, and coordination between, systems involved in memory and perception.

## Materials and methods

### Participants

Thirty individuals from the Columbia University community participated for monetary compensation ($12/hour for behavioral sessions and $20/hour for the fMRI session; $72 in total). The study was approved by the Institutional Review Board at Columbia University. Written informed consent was obtained from all participants. One participant did not perform well on the memory-guided attention task, as indicated by poor accuracy in using stay/switch cues to guide attention (M = 0.847). This person's accuracy was more than three standard deviations below the group average (M = 0.954; SD = 0.0317), suggesting that they were not effectively using memory to select attentional goals. We therefore excluded this participant from the analyses, leaving 29 participants (17 female; one left-handed; all normal or corrected-to-normal vision; 18–35 years old, M = 26, SD = 4.07; 13–21 years of education, M = 17.1, SD = 2.2).

### Design and procedure

#### Overview

There were two attentional states (art, room) and two tasks (memory-guided, explicitly instructed; *Figure 1*). In the 'art' attentional state, participants had to attend to the style of the painting in the base image (use of color, brushstrokes, level of detail) and determine whether any of the paintings in the search set could have been painted by the same artist who painted the painting in the base image (i.e., an art match: a painting that is similar in style). In the 'room' attentional state, participants had to attend to the layout of the room in the base image (arrangement of furniture, angles of the walls), and determine whether any of the rooms in the search set had the same spatial layout from a different perspective (i.e., a room match). Other aspects of the rooms (e.g., wall color, specific furniture exemplars) differed between the base image and its room match.

In the explicitly instructed task, the attentional state (art or room) was randomly assigned on each trial. In the memory-guided task, participants used memory for learned stay/switch cues to select their attentional goals: A stay cue on trial *N* indicated that the participant should stay in the same attentional state on trial *N*+1, while a switch cue on trial *N* indicated that the participant should switch to the other attentional state on trial *N*+1 (e.g., switch from 'room' to 'art' or from 'art' to 'room'). Finally, some trials contained neither a stay nor a switch cue. Following those 'no-cue' trials, participants were free to choose either 'art' or 'room' as their attentional state on the next trial.

Participants completed 4 runs of the memory-guided task and 4 runs of the explicitly instructed task (25 trials per run). All runs of the same type were completed before switching to the other task, and task order was counterbalanced across participants.

#### Stimuli

The images used in this study were 3D-rendered rooms, each of which contained one painting. The rooms were designed with Sweet Home 3D (sweethome3d.com). Each room contained multiple pieces of furniture and had a unique shape and layout. A second version of each room (to be used as its 'room match') was created with a 30° viewpoint rotation (half clockwise, half counterclockwise) and altered such that the content was different, but the spatial layout was the same. This was accomplished by changing the colors of the walls and replacing the furniture with different furniture of the same type at the same position (e.g., replacing a chair with another chair). The paintings were chosen from the Google Art Project. To obtain the 'art match' for each painting, a painting from the same artist was chosen, which had a similar style but whose content could differ. The combined images (art in a room) were generated by manually 'hanging' each painting along a wall.

2 paintings and 2 rooms were chosen to be 'stay' and 'switch' cues (1 painting and 1 room were 'stay' cues; 1 painting and 1 room were 'switch' cues). 12 'cue' images were generated by pairing each art cue (1 stay cue and 1 switch cue) with 3 different rooms, and each room cue (1 stay cue and 1 switch cue) with 3 different paintings. Thus, each stay/switch cue could appear in 3 different images. The 3 room 'backgrounds' for the art stay cue were the same as the 3 room 'backgrounds' for the art switch cue. Likewise, the 3 paintings embedded in the room stay cue were the same as the 3 paintings embedded in the room switch cue. The task-irrelevant portion of each stay/switch

cue (the room in art stay/switch cues and the art in room stay/switch cues) was therefore not diagnostic of the cue's identity.

The stimulus set used in the fMRI scan session contained 141 unique images (129 main images plus the 12 stay/switch cue images). These were derived from a set of 120 images (*Aly and Turk-Browne, 2016a*; *Aly and Turk-Browne, 2016b*) that were created by pairing each of 40 rooms with 3 different paintings (all by different artists) and each of 40 paintings with 3 different rooms (all with a different layout). We modified this set in order to pair the task-irrelevant feature of each stay/switch cue (e.g., the art in a room stay cue, or the room in an art stay cue) with 2 images used in the main stimulus set. That is, each of the 3 room 'backgrounds' for art stay/switch cues was also paired with 2 different paintings from the main stimulus set, and each of the 3 paintings embedded in room stay/switch cues was also paired with 2 different rooms from the main stimulus set. As a result, the task-irrelevant features of stay/switch cues was not diagnostic of the presence of these cues in any given trial. After these modifications, we had 129 main images comprising 43 rooms (40 main rooms plus three room 'backgrounds' from the art stay/switch cues) each paired with multiple paintings. Likewise, each of the 43 paintings (40 main paintings plus three paintings embedded in room stay/switch cues) were paired with multiple rooms.

20 images (unique art and room combinations) were chosen as 'base images.' These were used to create 20 'base sets' with 7 images each: a base image, a room match (an image with the same spatial layout as the base image, from a different perspective), an art match (an image with a painting by the same artist as the base image) and 4 distractors (rooms with different layouts and different artists compared to the base image). Room and art matches in one base set could be distractors in another base set. Base images were not used as distractors or matches in other base sets. An image that was an art match to the base image could not also be a room match to the base image, or vice versa. A given trial consisted of the presentation of a base image and 4 'search' images (from the pool of: art match, room match, distractors, stay/switch cue). Each base set was used to generate 10 trials: 5 in the memory-guided task and 5 in the explicitly instructed task.

A nonoverlapping set of 82 images (70 images plus 12 stay/switch cue images) were used during an initial practice day (~2 days before the fMRI scan). 70 main images were separated into 10 base sets of 7 images each (a base image, an art match, a room match, and 4 distractors). As in the scan session, 12 stay/switch cue images were generated by pairing each art cue (1 stay and 1 switch) with 3 different rooms, and each room cue (1 stay and 1 switch) with 3 different paintings. However, the stay/switch cues for this practice session were distinct from those used in the fMRI scan. The purpose of this session was to give individuals practice with the task, without exposing them to the specific stimuli to be used in the scanner.

An additional nonoverlapping set of 82 images (70 main images plus 12 stay/switch cue images) were used for a practice session that took place just before the fMRI scan. The 70 main images did not overlap with either the scan day images nor the initial practice day images (~2 days before the scan). The art and room stay/switch cues for this practice session were identical to those used during the fMRI scan. However, they were paired with rooms and paintings that were part of the 70 practice-specific images, which did not overlap with those used in the fMRI scan or the initial practice day. As in the other sessions, the art stay/switch cues were each paired with 3 rooms, and the room stay/switch cues were each paired with 3 paintings, making 12 stay/switch cue images in total.

## Design

Stimuli were presented using the Psychophysics Toolbox for MATLAB (psychtoolbox.org). At the beginning of each explicitly instructed trial, participants received the instruction to"Press any key with left hand to start the trial'. At the beginning of each memory-guided trial, participants received the instruction to 'Press left index for Room, left middle for Art'. This initiation screen remained visible until the participant responded. Apart from the initiation screen, the rest of the trial was identical for the explicitly instructed and memory-guided tasks.

After a key was pressed on the initiation screen in the explicitly instructed task, the attentional cue ('ART' or 'ROOM') was randomly assigned. In the memory-guided task, participants were instructed to select their attentional state based on the stay/switch cue in the preceding trial. This is similar to task-switching studies in which a cue (often an abstract one) signals when participants should switch to doing a different task (*Chiu and Yantis, 2009*; *Monsell, 2003*). For example, if the

attentional state on the previous trial was 'art', and there was an art 'switch' cue, then the attentional state on the current trial should be 'room' (art stay/switch cues only appeared on trials where art was attended; room stay/switch cues only appeared on trials where rooms were attended). If the participant mistakenly selected 'art', then the trial proceeded with an art attentional state. One-third of the trials did not contain a stay or switch cue. In the memory-guided task, following these 'no-cue' trials, and also on the first trial of each run, participants were free to choose whichever attentional state they wanted, but they were instructed to choose art and room approximately equally often. These no-cue trials were included in the design to test additional hypotheses beyond the focus of the present paper. Following these 'no-cue' trials, participants on average chose room (M = 16.828, 95% CI [16.299, 17.356]) more often than art (M = 14.655, 95% CI [14.144, 15.166]), $t(28) = 5.90$, $p<0.00001$, $d = 1.10$, 95% CI [1.418, 2.927]. However, this imbalance was only a few trials per participant (median = 3, min = 0, max = 7). Nevertheless, art and room trials were equally weighted in all analyses, so this slight difference could not account for any observed effects.

Following the initiation button press, participants were presented with the attentional cue, ('ART' or 'ROOM', centered at fixation), which remained on the screen for either 1.5 s, 2 s, or 2.5 s, randomized across trials. After the attentional cue, a base image was presented for 2 s. Then, four search images, centered at fixation, were presented for 1.25 s each, separated by 0.1 s inter-stimulus intervals. The 'ART?' or 'ROOM?' probe was then presented 0.1 s after the offset of the last search image, for a maximum of 2 s (less if the participant responded within that time). Participants indicated if there was a match present or absent by pressing the button box with the right-hand index or middle finger, respectively.

When the probe was 'ART?", participants' goal was to indicate if any of the paintings in the search images could have been painted by the same artist who painted the painting in the base image. For 'ROOM?' probes, participants' goal was to indicate if any of the room layouts in the search images was the same as that of the base image, but from a different perspective. 80% of trials were 'valid' trials, in which the attentional cue at the beginning of the trial matched the probe at the end. 20% of trials were 'invalid' trials, in which the attentional cue at the beginning of the trial did not match the probe at the end. This allowed us to ensure that attention was effectively engaged by the cue at the beginning of the trial (*Figure 2*).

Trials were separated by a blank inter-trial interval (ITI) of variable length. For each experimental run, the same set of 25 ITIs (truncated exponential, lambda = 1.5, mean = 6.66 s, T = 9 s) was used in a random order. At the end of each run, the percentage of correct responses was presented. In memory-guided runs, the accuracy of selecting the correct attentional state (based on the stay/switch cues) was also presented.

The same trial structure was used for the practice sessions except that the ITI was either 2 s or 2.5 s randomly determined on each trial. Furthermore, feedback on practice trials was shown after each probe (e.g., 'Correct, there was an art match'), and for the memory-guided task after each attentional state selection (e.g., 'Correct, there was an art switch cue on the previous trial').

Trial order was randomized with two constraints: (i) Each of the 20 base images was shown once every 20 trials, and (ii) the same base image was not repeated across two consecutive trials. The task-relevant match (e.g., an art match on a trial with an art probe) was shown on half of the trials, and, independently, the task-irrelevant match (e.g., a room match on a trial with an art probe) was shown on half of the trials. The remaining images in the search set were distractors, chosen among the four distractor image options for a given base set. A given image was never shown twice in a trial. On two-thirds of the trials, one of the 12 stay/switch cue images replaced one of the distractor images (this was true for both the memory-guided and explicitly instructed tasks; for the explicitly instructed task, these stay/switch cues had no relevance for the attentional cue on the following trial).

Valid trials of each task (memory-guided, explicitly instructed) were distributed across the two attentional states (art; room), two task-relevant match types (match present; match absent), two task-irrelevant match types (match present; match absent), and three cue types (stay; switch; none) as equally as possible. Although perfectly equating trial numbers across conditions was not possible for a given participant, trial numbers were equated every six participants.

## Procedure

Participants first came in for a behavioral practice session approximately 2 days before the fMRI scan. This session involved training in both the memory-guided and explicitly instructed tasks, but with stimuli that were non-overlapping with those used in the fMRI session. Both the practice session and the fMRI session followed the same procedure (below).

On each practice session (~2 days before the fMRI scan and on the day of the fMRI scan), participants completed 3 phases of practice. First, they completed a run of 10 trials in the explicitly instructed task. This run was repeated until participants reached at least 65% accuracy on validly cued trials. Next, participants completed the stay/switch cue learning phase. Here, the stay/switch cue images and their meanings (i.e., stay or switch) were presented for four times each in shuffled order, for a minimum of 1 s. The participant had to push a button to continue to the next image. Then, the stay/switch cue images were shown again, this time without their meanings (i.e., no stay/switch label), five times each in shuffled order. Participants indicated if a given image was a stay or switch cue with a button press. Completion of this phase required responding accurately to every image 5 times in a row. Upon a single incorrect response, this test phase was terminated, and the stay/switch cue learning phase was restarted from the beginning by presenting each stay/switch cue image and its meaning for 4 times. After completing the stay/switch cue test, participants performed a run of 10 trials in the memory-guided task. This memory-guided practice session ended once participants reached, in a given run of 10 trials, a minimum of 65% accuracy for validly cued trials in the attention task and a minimum of 80% accuracy for selecting the correct attentional state based on stay/switch cues.

Participants then completed the fMRI task, for which there were 8 runs of 25 trials each. Explicitly instructed (100 trials) and memory-guided (100 trials) tasks were blocked to constitute either the first or second half of the experimental session (order counterbalanced across participants). When starting a new task, participants performed five practice trials to get used to that particular task. The practice was repeated until accuracy on the art/room attention task was at least 65%. In memory-guided runs, the practice was also repeated until accuracy in selecting the appropriate attentional state based on stay/switch cues was at least 80%. At the end of each memory-guided task run, participants were shown a reminder screen with all four stay/switch cues (two paintings, two rooms) and their meanings (i.e., stay or switch). If on a given memory-guided task run, the average accuracy of choosing the correct attentional state was less than 85%, then the stay/switch cue learning phase (mentioned in the previous paragraph) was repeated.

Our design has several important aspects. First, a room match and an art match were equally and independently likely to be present in search images, for both art trials and room trials. Thus, accurate responding required being in the correct attentional state. Second, the same stimuli were used for the art and room attentional states (except for the stay/switch cues), so that differences in brain activity for these states must reflect top-down attentional goals rather than differences in the stimuli presented. Third, stimuli were identical across the memory-guided and explicitly instructed tasks, including the stay/switch cues. However, in the explicitly instructed task, the presence of a stay or switch cue did not have any consequence for participants' attentional states (because these states were randomly assigned on each trial). Thus, differences in brain activity between the memory-guided and explicitly instructed tasks cannot be due to the mere presence of stay/switch cues, but rather must be due to the need to use these cues to guide attention. Finally, motor demands were the same for the memory-guided and explicitly instructed tasks. Thus, the only difference between these tasks was the need to use memory to guide attention.

## MRI acquisition

MRI data were collected on a 3 T Siemens Magnetom Prisma scanner with a 64-channel head coil. Functional images were obtained with a multiband echo-planar imaging (EPI) sequence (repetition time = 1.5 s, echo time = 30 ms, flip angle = 65°, acceleration factor = 3, voxel size = 2 mm iso), with 69 oblique axial slices (14° transverse to coronal) acquired in an interleaved order. There were eight functional runs, four for the explicitly instructed task and four for the memory-guided task. Whole-brain high-resolution (1.0 mm iso) T1-weighted structural images were acquired with a magnetization-prepared rapid acquisition gradient-echo sequence (MPRAGE). Field maps were collected to aid registration, consisting of 69 oblique axial slices (2 mm isotropic).

## fMRI analysis

### Software

Preprocessing and analyses were performed using FEAT, FNIRT, and command-line functions in FSL (e.g., fslmaths). ROI (region of interest) analyses (e.g., univariate activity, pattern similarity, and multivariate connectivity) were performed using custom Matlab scripts. Data, experiment code, and analysis code are publicly available on the Open Science Framework: https://osf.io/ndf6b/.

### ROI definition

The hippocampus ROI was anatomically defined from the Harvard-Oxford atlas in FSL (*Jenkinson et al., 2012*). The vmPFC ROI was based on *Mackey and Petrides (2014)*, but we removed voxels that overlapped with the corpus callosum. The V1-2 ROI was obtained from the human visual cortex atlas provided in *Wang et al. (2015)*. ROIs are shown in *Figure 7*.

### Preprocessing

The first 4 volumes of each run were discarded to allow for T1 equilibration (except for one participant for whom only one extra volume, rather than 4, was collected for this reason). Brain extraction, motion correction (using the MCFLIRT motion correction tool of FSL; *Jenkinson et al., 2002*), high-pass filtering (cut-off = 128 s), and spatial smoothing (3 mm FWHM Gaussian kernel) were performed as preprocessing steps. Field map preprocessing was based on recommendations in the FUGUE user guide (https://fsl.fmrib.ox.ac.uk/fsl/fslwiki/FUGUE/Guide) and carried out with a custom script. First, two magnitude images were averaged and skull stripped. The average magnitude image was then used together with the phase image to generate a field map image using the fsl_prepare_fieldmap command of FSL. This field map image and the average magnitude image were included in the preprocessing step of FEAT analyses to unwarp the functional images and aid registration to anatomical space. This approach helped to reduce the distortion in anterior temporal and frontal regions. Functional images were registered to the standard MNI152 T1-weighted structural image using a non-linear warp with a resolution of 10 mm and 12 degrees of freedom.

### Image period — Univariate Activity

Only valid trials were used for image period analyses, to reduce any potential BOLD signal contamination from an invalid probe (as in *Aly and Turk-Browne, 2016a*; *Aly and Turk-Browne, 2016b*; note, however, that our results hold when all trials are used). We included trials with both correct and incorrect responses to balance the number of trials per participant. To test if univariate activity levels were higher for memory-guided vs. explicitly instructed tasks, we modeled the data with a single-trial GLM. Each trial (25 in each run) was modeled as a 7.4 s epoch from the onset of the base image to the offset of the last search image. There were two additional regressors: a regressor for all orienting periods, modeled as the interval from the onset of the initiation screen (which remained until a key was pressed) until the offset of the attentional cue; and a regressor for all probe periods, modeled as a 2 s epoch during the probe display. All regressors were convolved with a double-gamma hemodynamic response function. Finally, the 6 directions of head motion were included as nuisance regressors. Autocorrelations in the time series were corrected with FILM prewhitening. Each run was modeled separately, resulting in eight different models per participant. For each participant, image period parameter estimates were averaged across voxels within each ROI, and the resulting values for the memory-guided and explicitly instructed tasks were compared at the group level using a paired-samples t-test.

To test if the activity enhancement for memory-guided vs. explicitly instructed tasks was correlated between the hippocampus and vmPFC across individuals, we first subtracted the average parameter estimate in the explicitly instructed task from that of the memory-guided task for each individual, separately for the hippocampus and vmPFC. Then, these memory-guided vs. explicitly instructed difference scores in the hippocampus and vmPFC were correlated across individuals using the skipped_correlation.m function (https://github.com/CPernet/robustcorrtool; *Pernet et al., 2012*). This function performs a robust correlation by removing bivariate outliers as determined by: (1) finding the central point in the distribution using the minimum covariance determinant (*Rousseeuw and Driessen, 1999*), (2) orthogonally projecting each data point onto lines that join each data point to the estimated center point, (3) identifying outliers on the projected data using

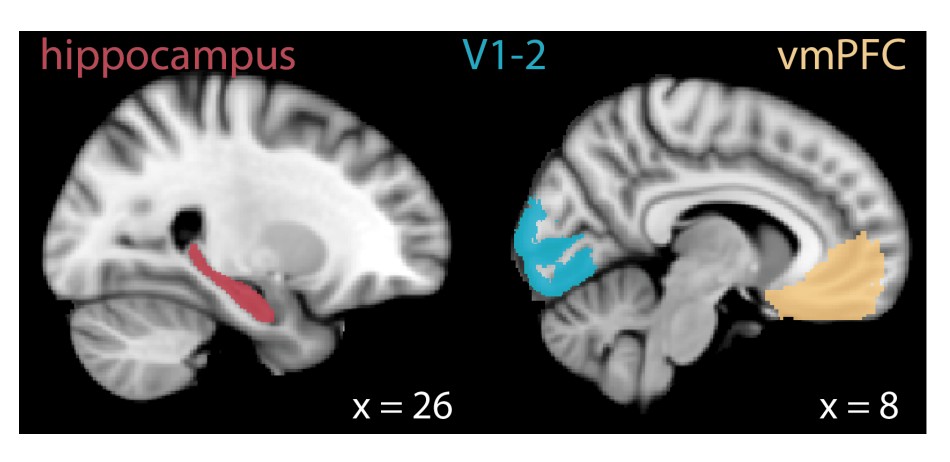

**Figure 7.** Regions of interest. Hippocampus (red), V1-2 (blue), and vmPFC (orange) are shown in the right hemisphere of the brain, but all regions of interest are bilateral.

the interquartile range method (*Frigge et al., 1989*), and (4) calculating Pearson's correlation after removing the outlier(s). With this approach, one participant was excluded as an outlier. However, our results hold when this participant was included in the analysis ($R^2$ = 0.44, p=0.000092, 95% CI [0.39, 0.83]).

## Image period — Pattern Similarity

To test if multivariate patterns of activity represent online attentional goals, we conducted pattern similarity analysis on trial-specific activity patterns from the image periods. This, and all other multivariate analyses, were conducted on preprocessed fMRI data. First, preprocessed data were z-scored across time, within each voxel, separately for each run. Data were then shifted forward by 6 s (4 TRs) to account for hemodynamic lag before selecting TRs that corresponded to each image period. Specifically, TRs for which at least half of the volume acquisition time corresponded to the (shifted) image period were considered to be image period TRs. Mean activity patterns for each image period were obtained for each region of interest by averaging activity levels for each voxel across all image period TRs. These activity patterns were then reshaped into a single-dimensional vector (length = number of voxels). Pearson correlations were then calculated between all pairs of vectors (i.e., between all trials) in different runs of the same task (i.e., task-specific pattern similarity was obtained by comparing explicitly instructed runs to other runs of the same task, and memory-guided runs to other runs of the same task). Correlations between trial pairs within the same run were excluded to reduce the effects of autocorrelation on pattern similarity values (*Mumford et al., 2014*). Finally, correlations were averaged separately for same state trial pairs (i.e., art-art, room-room) and different state trial pairs (i.e., art-room). For statistical testing, correlations were Fisher-transformed before averaging to ensure normality. Fisher-transformed pattern similarity values for trials of the same vs. different attentional states were compared at the group level with a paired-samples t-test, separately for the memory-guided and explicitly instructed conditions.

We used only valid trials for the image period analysis — as in our previous work (*Aly and Turk-Browne, 2016a*; *Aly and Turk-Browne, 2016b*) — because neural activity on invalid trials might partly reflect the invalidly probed attentional state representation. However, we obtained the same pattern of results when both invalid and valid trials were included in the analysis.

## Image period — Multivariate Connectivity

To examine interactions between the hippocampus and early visual cortex, we calculated multivariate connectivity between these regions. In order to do this, we first obtained measures of the 'quality' of attentional states in each region on a trial-by-trial basis, and then correlated these attentional state quality measures between regions.

For this analysis, we used the z-scored, preprocessed data as we did for the image period pattern similarity analysis mentioned above. First, we averaged activity patterns across trials, separately for art and room attentional states, to create art and room attentional state 'templates'. These templates indicate what brain activity in a given region generally looks like for the art vs. room attentional states. Second, we calculated Pearson correlations between these templates and activity patterns for individual trials. Importantly, the templates compared to a given trial excluded trials in the same run (e.g., for analysis of a trial in run 1, templates excluded other trials in run 1; *Mumford et al., 2014*). Third, for each trial's activity pattern, we calculated a measure of multivariate attentional state 'quality' by subtracting its correlation with the different-state template (e.g., an art trial correlated with the room template) from its correlation with the same-state template (e.g., an art trial correlated with the art template). These steps were performed separately for the hippocampus and V1-2. Lastly, we computed multivariate connectivity between the hippocampus and V1-2 by calculating Pearson correlations for their multivariate attentional state 'quality' scores across all trials. For statistical testing, multivariate connectivity values were Fisher-transformed to ensure normality. The Fisher-transformed connectivity values for the explicitly instructed and memory-guided tasks were compared at the group level using a paired-samples t-test. These values were also compared to zero using a one-sample t-test for each task.

## Orienting period — Univariate Activity

To examine whether preparatory univariate activity in the hippocampus was enhanced for memory-guided vs. explicitly instructed attention (*Stokes et al., 2012*), we examined BOLD activity in the hippocampus during the orienting period. To this end, we performed a single-trial GLM with 27 regressors. There were 25 orienting period regressors (one for each orienting period), modeled from the onset of the initiation screen (which remained until a key was pressed) until the offset of the attentional cue; a single regressor for all image periods, modeled as 7.4 s epochs from the onset of the base image to the offset of last search image; and a regressor for all probe periods, modeled as 2 s epochs during the probe displays. As in the image period analyses, (i) all regressors were convolved with a double-gamma hemodynamic response function, (ii) the 6 directions of head motion were included as nuisance regressors, (iii) autocorrelations in the time series were corrected with FILM prewhitening, (iv) both correct and incorrect responses were included, (v) only valid trials were used (our results hold when invalid trials are included), and (vi) each run was modeled separately. The first trial of each run was excluded from the orienting period analysis, as there was no previous trial for the attentional state decision to be based on (in the memory-guided task). For each participant, orienting period parameter estimates were averaged across voxels, and the resulting values for the memory-guided and explicitly instructed tasks were compared at the group level using a paired-samples t-test. For completeness, we also performed this analysis for vmPFC.

## Orienting period — Pattern Similarity

To test if multivariate activity patterns during the orienting period represented preparatory attentional states, activity patterns during the orienting periods were correlated with activity patterns from the image periods. As in the image period analysis, we used preprocessed and z-scored data.

Given that we were interested in the correlation between the activity patterns of these two temporally adjacent periods, we attempted to limit their autocorrelation — induced by the slow hemodynamic response — in two ways. First, we only compared orienting period activity patterns and image period activity patterns across runs, i.e., the orienting period activity patterns on run 1 were never compared to image period activity patterns in run 1 (*Mumford et al., 2014*). Second, to further reduce their autocorrelation, we removed boundary TRs from the analysis (i.e., the last TR of the orienting period and the first and last TR of the image period; it was not necessary to drop the first TR of the orienting period because it followed a blank inter-trial interval). The first TR of the image period was removed to reduce autocorrelation with the orienting period, which is important given that we used the correlation between these two periods as evidence for preparatory attentional states. The last TR of the image period was not included so as to remove BOLD activity due to the probe. This is particularly critical when comparing orienting period activity patterns to image period activity patterns as a marker of preparatory attentional states: because the cue component of the orienting period ('ART' or 'ROOM') overlaps perceptually with the probe ('ART?' or 'ROOM'?),

not dropping the last TR of the image period risks an artificial boost of orienting period/image period pattern similarity as a result of this perceptual overlap. Note that a TR was considered to be part of the image period if at least 50% of the duration of that brain volume acquisition corresponded to the image period, but this still leaves a considerable amount of time for the probe to affect brain activity during that TR. For these reasons, the last TR of the image period was dropped in order to be conservative.

Importantly, the last TR of the orienting period was removed only if it was not the only TR during which the attentional cue was presented. This ensured that orienting period activity patterns always included timepoints at which the attentional state was known to the participant. This is particularly important for the explicitly instructed condition: Otherwise, a difference between the memory-guided and explicitly instructed tasks could simply arise because participants know their attentional state in one task but not in the other. Thus, this step ensured that any differences between tasks are because of *how* attentional state information was obtained (from memory or an overt instruction), rather than its availability.

After dropping boundary TRs in this way, we obtained a mean activity pattern for each period (orienting or image) by averaging over the remaining TRs. Because we dropped boundary TRs in this analysis, but not in the main image period analysis, we first confirmed that the hippocampus and vmPFC still discriminate between the art and room attentional states during the image period with this new, conservative approach. Indeed, both the hippocampus (memory-guided: $t(28) = 2.87$, p=0.00078, $d = 0.53$, 95% CI [0.001, 0.007], explicitly instructed: $t(28) = 6.59$, p<0.0001, $d = 1.22$, 95% CI [0.007, 0.013]) and vmPFC (memory-guided: $t(28) = 6.61$, p<0.0001, $d = 1.23$, 95% CI [0.009, 0.017], explicitly instructed: $t(28) = 6.06$, p<0.0001, $d = 1.13$, 95% CI [0.011, 0.023]) still exhibited greater pattern similarity for trials of the same vs. different attentional states in both tasks.

Having confirmed distinct representations for the art and room attentional states with this approach, we next obtained 'template' activity patterns for the *image periods*. These template activity patterns were the average of valid art trials (for the art template) and the average of valid room trials (for the room template). The purpose of these templates was to obtain activity patterns that represent *online* attention to artistic styles vs. room layouts.

Activity patterns for the *orienting period* of each trial were then correlated with the art template and room template. Importantly, the image period templates excluded all trials in the same run as a given orienting period activity pattern (e.g., for the analysis of the orienting period in run 1, image period templates excluded trials in run 1). The correlations between the orienting period activity patterns and the image period templates were then grouped based on whether they were a match to the same state (e.g., an art trial orienting period activity pattern correlated with an art image period template) or a match to the different state (e.g., an art trial orienting period activity pattern correlated with a room image period template). This was repeated for all trials. The correlations were then averaged separately for each combination of match type (match to same template; match to different template), attentional state (art; room), and task (explicitly instructed; memory-guided).

These pattern similarity values were averaged across attentional states (art, room). This ensured that art and room trials contributed to average pattern similarity values equally. To measure preparatory attentional states, we calculated the difference in average pattern similarity for match-to-same template and match-to-different template correlations, separately for each participant. The match-to-same template and match-to-different template difference scores were then compared for the memory-guided and explicitly instructed tasks with a paired-samples t-test, after Fisher-transforming these values to ensure normality. Finally, the difference scores were compared to zero using one-sample t-tests, separately for memory-guided and explicitly instructed tasks. Values significantly above 0 indicate evidence for the upcoming attentional state.

## Orienting period — Whole-Brain Searchlight

To test whether other brain regions represent preparatory attentional states, we performed the orienting period pattern similarity analysis using a whole-brain searchlight approach, via the Simitar toolbox (*Pereira and Botvinick, 2013*). This analysis was identical to the main orienting period ROI analysis (*Figure 1—figure supplement 1*) except that pattern similarity was calculated for all possible 27-voxel cubes (3×3×3 voxels) throughout the brain. The result (i.e., orienting period to image period pattern similarity) for each cube was assigned to the center voxel. This analysis was

conducted separately for each participant, and group-level statistics were then performed with the randomise function in FSL. Specifically, we performed a non-parametric one-sample t-test that used 10,000 permutations to generate a null distribution. Voxel-based thresholding was applied, corrected for multiple comparisons using the family-wise error rate correction (p<0.05).

## Acknowledgements

We would like to thank Trevor Dines, Daniella Garcia-Rosales, Andrew Goulian, Bobby Hickson, Caroline Lee, Tamar Mosulishvili, Alexandra Reblando, Nicholas Ruiz, and Debby Song for help with data collection; Dania Elder, Ray Lee, and Julie Kabil for their technical help with the MR scanner; and Lila Davachi and her lab for advice on the project design. This work was funded by an NSF CAREER Award (BCS-1844241) and a Zuckerman Institute Seed Grant for MR Studies (CU-ZI-MR-S-0001) to MA.

## Additional information

### Funding

| Funder | Grant reference number | Author |
| --- | --- | --- |
| National Science Foundation | BCS-184421 | Mariam Aly |
| Zuckerman Institute | Seed Grant for MR Studies (CU-ZI-MR-S-0001) | Mariam Aly |

The funders had no role in study design, data collection and interpretation, or the decision to submit the work for publication.

### Author contributions

Eren Günseli, Conceptualization, Formal analysis, Investigation, Methodology, Writing - original draft, Writing - review and editing; Mariam Aly, Conceptualization, Formal analysis, Supervision, Funding acquisition, Investigation, Methodology, Writing - original draft, Writing - review and editing

### Author ORCIDs

Eren Günseli (iD) https://orcid.org/0000-0002-7944-7774
Mariam Aly (iD) https://orcid.org/0000-0003-4033-6134

### Ethics

Human subjects: The study was approved by the Institutional Review Board at Columbia University (Protocol number: AAAR5338). Written informed consent was obtained from all participants.

### Decision letter and Author response

Decision letter https://doi.org/10.7554/eLife.53191.sa1
Author response https://doi.org/10.7554/eLife.53191.sa2

## Additional files

### Supplementary files

• Transparent reporting form

### Data availability

All data used in the analyses is publicly available on Open Science Framework: https://osf.io/ndf6b/.

The following dataset was generated:

| | Database and |
| --- | --- |

| Author(s) | Year | Dataset title | Dataset URL | Identifier |
|-----------|------|---------------|-------------|------------|
| Günseli E, Aly M | 2019 | Aly Lab: Preparation for upcoming attentional states in the hippocampus and medial prefrontal cortex | https://osf.io/ndf6b/ | Open Science Framework, ndf6b |

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
