## [Decision Letter]

**Acceptance summary:**

This work elegantly demonstrates that memories flexibly guide attention by enabling preparation for upcoming representational states in the hippocampus and medial prefrontal cortex. This behaviour requires close coordination between the hippocampus and early visual cortex, indicating the inextricable relationship between perception, memory, and attention.

**Decision letter after peer review:**

Thank you for submitting your article "Preparation for upcoming attentional states in the hippocampus and medial prefrontal cortex" for consideration by *eLife*. Your article has been reviewed by two peer reviewers, and the evaluation has been overseen by a Reviewing Editor and Timothy Behrens as the Senior Editor. The following individual involved in review of your submission has agreed to reveal their identity: Kia Nobre (Reviewer #1).

The reviewers have discussed the reviews with one another and the Reviewing Editor has drafted this decision to help you prepare a revised submission.

Günseli and Aly use fMRI to investigate the mechanisms by which memories guide attention using two tasks – one in which attention guided memory and one in which attention was explicitly instructed. In a univariate analysis, they found that activity levels in both the hippocampus and vmPFC were higher for memory-guided versus explicitly-instructed attention. Using representational similarity analysis, they found that activity in the hippocampus and vmPFC contained information about current and upcoming attentional states, but that in the hippocampus preparatory attentional state representations were stronger for memory-guided versus explicitly instructed attention. The hippocampus and visual cortex showed increased synchrony in their attentional state representations in the memory-guided versus explicitly-instructed task, suggesting that the role of the hippocampus in memory-guided attention is supported by its interactions with visual cortex.

The reviewers found the question interesting, the study well motivated, and the findings robust. Although they expressed enthusiasm for the value for the work, the reviewers converged on three major issues, which I have broadly summarized below:

1) The primary concern voiced by both reviewers was the operationalization of the decision period and the fact that there is that there is no clean way to isolate the preparatory task set without running a new experiment with other parameters. The reviewers are not requesting that this experiment be run, but they do believe that a fulsome discussion of limitations and a reworking of some of the interpretations is necessary.

2) Both reviewers commented on the paper's primary framing around attention and the fact that the attentional manipulation used was not necessarily straightforward. In particular, the reviewers felt that the fact that the manipulation is to the task set rather than to any content/attribute of the stimulation (e.g., spatial location, object, feature, timing) should be discussed and addressed very carefully. It was noted, however, that the paper describes strong behavioural effects for cueing/prompting of the task set, and thus, can be considered to be a type of high-level attention manipulation.

3) Both reviewers highlighted some concerns and comments regarding task/condition difficulty differences and how they impacted the analyses.

Each reviewer also provided some additional concerns and suggestions, which can be found in their specific individual reviews. I hope that these are helpful to you as you prepare your revision.

Reviewer #1:

Günseli and Aly conducted an fMRI study using representational similarity analysis to investigate the involvement of the hippocampus and the mPFC in memory-guided attention. The question is very interesting and the study was well motivated. Although their experimental manipulation of memory-guided attention is unorthodox, their findings are robust and interesting. The study helps highlight the importance of considering how memories of different time scales contribute to proactive attention.

1) The specific memory-guided attention manipulation was unorthodox. In previous studies, learned associations guided spatial attention on the basis of the content of memories. Here, learned associations are used to signal the dimension to which participants must attend (art or room), without predicting any of the contents to be anticipated. This is like learning the meaning of an explicit attention cues directing individuals to switch or stay with their current attentional focus (as in studies by Yantis).

In addition, the manipulation brings with it an element of endogenous, self-directed choice of what to attend supported by relatively short-term memories of what they encountered on a previous trial. The manipulation brings significant additional demands to the task – e.g., WM, multiplexing task at hand and goal setting, choice. A similar manipulation in spatial perceptual attention tasks was performed by Taylor, Rushworth and Nobre, 2008.

The authors should consider and discuss how the particularities of their design will have affected their findings relative to how memory-guided attention has been studied in the past.

2) The authors are interested in predictive anticipatory states, but there are no long periods during the task in which stimulation is absent for deriving clean anticipatory neural states. The 'decision period' is short and insufficiently separated from the image period.

(Calling it a decision period is also confusing, since that would typically relate to the decision at the probe phase. Perhaps the 'orienting' period?)

The authors address this issue (subsection “Robustness of preparatory attentional states”), but the limitation in the design means the issue remains inconclusive.

3) Given the additional demands of the memory task, could these have contributed to univariate differences in HC and mPFC activations between the task conditions? How can one rule this out?

4) Given the nature of the attention manipulations, the multivariate effects were not linked to any content in the images, but to the dimension of the images (art, room) that was relevant. Could patterns also reflect nuisance factors other than focus on the information relevant to the goal? For example, some participants may have found one of the dimensions more difficult and therefore modulated arousal or effort levels to compensate and achieve similar levels of performance?

5) Separating retrieval of a task set from preparation to its utilisation is conceptually very difficult and may not be possible in practise. Without some content-related expectation that is separate from the remembered/cued task set, this topic cannot really be addressed in the current experimental design. In previous studies (e.g., Stokes, 2012), memory for an association predicted the specific location of a target, so that differential levels of activity in visual cortex could be compared accordingly.

(However, to some extent, this distinction may not always be useful, and one might instead consider how an act of retrieval can itself change the state of the system in a way that changes how it processes incoming information.)

6) Although the focus on HC and mPFC is well motivated and sensible, is there a risk that the authors miss where the real action is? It would be good to provide a supplementary exploratory analysis (using appropriately strict corrections for multiple comparisons) to reassure readers that the study captures the main players in in the cognitive functions examined and/or to highlight any additional important brain area/relationship that could be pursued in future studies.

Reviewer #2:

The manuscript describes an fMRI study that used a room/artist task to investigate task set ("attentional states") representation in hippocampus and vmPFC, during task cue and task performance periods. Task representations observed during task performance generalized to the task cue period, indicating they are abstract in nature. Univariate analyses further indicated that hippocampus and vmPFC are more strongly engaged when the information about which task (room or art) needs to be performed is memory-based than externally cued.

1) This was a thorough and well written report of a variation on the artist/room task. My first comment, probably apparent from my summary, is that I am not entirely on board with the whole framing around attention or "attentional states". The terminology has been used by the authors previously, but I find it somewhat idiosyncratic. The way I would write about this same experiment: there are two conditions, memory-guided vs. explicitly instructed. Participants are performing one of two tasks, artist or room. We are looking at how the task is represented when cued or being performed. Yes, the task determines what participants should be paying attention to, but that does not necessarily mean that what is being decoded by RSA is best described as "attentional state". Something like "task representation" seems more neutral, and it would align with terminology commonly used in task-switching literature that looks at very similar questions. Even in memory, we have studies that use living/non-living judgment on some trials and bigger/smaller than shoe box on other trials. They are still described as two tasks (not attentional states) and have been also analyzed similarly to look at task (or "task set") representation. I am not opposed for the authors to put forth their preferred interpretation in the Discussion. But I find the term "attentional state" over-interpretative and potentially misleading when used as the main framing. Also, it was never defined.

2) Univariate analyses (Figure 3): More control needs be done for task/condition difficulty differences. Memory vs. external condition behavioral performance difference was *p* = 0.084 (subsection “Behavior”), memory vs. external condition hippocampus effect was *p* = 0.011. As the former-behavior-was dismissed as no difference, it was not considered as a possible source of the latter. However, control analyses should be conducted, taking the marginal behavioral differences into account. This holds for results from both Figure 3A and Figure 3B.

3) Univariate results interpretation. Even if activation difference in Figure 3 hold after controlling for behavioral differences, what they mean may be different from what is proposed. The results are from the image period, where "attentional state" (room or artist task) is no longer memory-guided. The “ART” or “ROOM” cue was explicitly presented once the memory-guided decision was made earlier in the trial. By the image presentation period, the cue became external in both conditions, making them comparable in that regard.

What differentiates the conditions instead is that the memory-guided condition is a dual-task condition. Participants need to conduct the room or art task AND ALSO keep track of a potential switch/stay cues (that needed to be memorized). This offers a simple explanation of any activation differences between the memory-guided and explicitly instructed condition.

4) Operationalization of the "decision period". I did not have any issues with the RSA analysis per se, and appreciated the authors' thorough control analyses. However, I did not find it appropriate to call the beginning of the trial as "decision period" for two reasons. (1). There wasn't a task-relevant decision to be made in the explicitly-instructed period. (2) The art/room cue period was included. Given their proximity, there is no way to get rid of the art/room cue-related activation from this analysis. My view of this analysis and the results is that we are primarily measuring cue-related task representation (room/art). For example, both hippocampus and vmPFC represent “ROOM” vs. “ART” above chance in the explicitly instructed condition, which clearly does not come from the decision whether to press index or middle finger. Thus, the term "decision period", which lends itself to implying decision-related activity, is misleading in the current context.

5a) Additional concerns related to the interpretation of the multivariate results: As noted above, most straightforward interpretation of Figure 5 does not involve "decision" per se. It seems to be about room/art task set representation ("which task I'm doing"?). The match between the cue period and image period indicates that the room/art task representation is abstract in nature (not the specific task cues "ART" or "ROOM"). Whether it is appropriate to call that preparatory attentional signal is not apparent. Of course, participants need to remember which task they are doing throughout the image period. Thus, one may also see the relationship the other way – decoding during image period reflects memory for the cue from the beginning of the trial.

5b). In addition, the authors found stronger representation in memory-guided than explicitly instructed condition in the hippocampus. A challenge with interpreting this result is different duration of the event of interest under the two conditions. In the memory-guided condition, it is already apparent during the decision screen (and on the majority of trials, even before the decision screen during ITI) whether the next trial is the room task or the art task. Thus, brain activation during both decision and task-cue screens (modeled together here) can reflect the room/art task identity. In contrast, in explicit condition, the room vs. art task information is not available until the task-cue screen, and thus the brain cannot possibly reflect room vs. art task identity while the subject is deciding whether they want to press their index or middle finger. Perhaps I misunderstood the procedures. But if not, I don't see the differences between art vs. room decoding between the conditions as interpretable.

---

## [Author Response]

1) The primary concern voiced by both reviewers was the operationalization of the decision period and the fact that there is that there is no clean way to isolate the preparatory task set without running a new experiment with other parameters. The reviewers are not requesting that this experiment be run, but they do believe that a fulsome discussion of limitations and a reworking of some of the interpretations is necessary.

We agree with the reviewers’ assessments. Below are some of the main changes, and further changes are discussed in response to individual reviewer comments.

a) The decision period is now called the ‘orienting period’, given the comments by both reviewers that the previous terminology was confusing and inaccurate. The new terminology was chosen based on a recommendation from reviewer #1.

b) We now address the limitations of the current approach in the Discussion. In the Results section of the initial manuscript, we discussed why we think the preparatory signals we observed are not due to autocorrelation in the BOLD signal but we agree that this consideration should also be raised as a main limitation in the Discussion. We have therefore added the following text in the subsection entitled “Nature of preparatory attentional states”. The new text also discusses a number of alternative explanations for the content of preparatory signals. These were raised by the reviewers and we agree that they are feasible.

“When a brain region prepares for, or anticipates, an upcoming task, what is being represented? […] This would be particularly useful if fMRI were complemented with EEG, to incorporate the high temporal resolution of the latter method (e.g., Stokes et al., 2012).”

We also hope that the care we took to discuss the issue of autocorrelation, in *Robustness of preparatory attentional states*, is helpful – but will be happy to take further advice if not.

c) We discuss whether comparing preparatory attentional states across the memory-guided and explicitly instructed conditions is problematic and how future studies can improve on our design. Here, we also mention the possibility that, due to the relatively short duration of the orienting period – and the lack of a delay between it and the image period – the hippocampus might not have had enough time to anticipate upcoming attentional goals in the explicitly instructed task. As a result, the stronger preparatory representations in the hippocampus for the memory-guided task may not be observed in studies that have a longer blank delay between the orienting period and the image period.

“The current study confirmed our hypothesis that the hippocampus and vmPFC prepare for upcoming attentional states. […] Such methods can establish the temporal dynamics by which memory-guided vs. explicitly instructed attention influence representations across different brain regions.”

2) Both reviewers commented on the paper's primary framing around attention and the fact that the attentional manipulation used was not necessarily straightforward. In particular, the reviewers felt that the fact that the manipulation is to the task set rather than to any content/attribute of the stimulation (e.g., spatial location, object, feature, timing) should be discussed and addressed very carefully. It was noted, however, that the paper describes strong behavioural effects for cueing/prompting of the task set, and thus, can be considered to be a type of high-level attention manipulation.

We agree that there are numerous differences between our manipulation of attention and those typically used in other studies – particularly studies of memory-guided attention from which we drew inspiration. We have addressed this in multiple ways. The main changes are noted below, and additional changes are discussed in response to individual reviewer comments.

a) We define what we mean by ‘attentional state’. We did not do this before, and that was certainly a shortcoming. In our definition, we highlight the similarity to task representations or task sets.

“Here, we examine the mechanisms underlying memory-guided attention with the aim of determining the nature of neural representations that enable past experiences to be used to prepare for upcoming attentional states. […] Attentional states can be considered an instance of a task representation or a task set (Mayr and Kliegl, 2000; Sakai, 2008), with the task defining what should be attended to.”.

b) We discuss the many ways that our manipulation differs from more ‘standard’ attention manipulations. At the same time, we highlight the robust behavioral effects that are consistent with our conceptualization of this as an attentional manipulation:

“Our work was inspired by studies of memory-guided attention (e.g., Stokes et al., 2012; Summerfield et al., 2006) but it differs from them in a number of ways. […] Second, our study converges with other studies of memory-guided attention in suggesting that the hippocampus plays a role in guiding attentional behaviors on the basis of past experience (see Aly and Turk-Browne, 2017, for review).”

c) We mention procedural differences when comparing our results to other studies of memory-guided attention:

“We also found that these regions showed no difference in univariate activity levels between the memory-guided and explicitly instructed conditions during the orienting period. […] Thus, differences in the kind of information carried by memory (specific content vs. abstract task set), as well as in the timing of the orienting periods and the attention task, could have led to the observed differences in univariate activity during preparatory attention.”

3) Both reviewers highlighted some concerns and comments regarding task/condition difficulty differences and how they impacted the analyses.

We have run the requested analyses to address whether differences in task difficulty could account for any observed results. These analyses are summarized here and discussed more fully below, in response to the individual reviewer comments.

a) We found that the univariate activity enhancement for memory-guided vs. explicitly instructed attention was not related to differences in performance across the two tasks. This was the case for both hippocampus and vmPFC.

“To determine if this difference in univariate activity is related to differences in behavioral performance across tasks, we examined whether A’ differences on the memory-guided vs. explicitly instructed task predicted univariate activity differences between these two tasks, across individuals. […] Thus, univariate activity enhancement in these regions for memory-guided attention cannot be explained solely by differences in behavioral performance.”

b) We found that the correlation between hippocampal and vmPFC activity enhancements for memory-guided vs. explicitly instructed attention remained (and was largely unaffected) when controlling for individual differences in behavioral performance across the memory-guided and explicitly instructed tasks.

“If the hippocampus and vmPFC work together to establish memory-guided attentional states, then the extent to which one region’s activity is modulated by memory-guided attention might predict how much the other region’s activity shows such modulation. […] In the Discussion, we further consider what enhanced univariate activity in these regions might reflect.”

c) We found that individual differences in performance across the art vs. room attentional states did not correlate with the strength of preparatory attentional signals. This helps address the concern that individuals may have simply been modulating arousal or effort in anticipation of a challenging task.

“One possibility is that preparatory attentional states observed in our study reflect the anticipated difficulty of art and room attentional states. […] That said, differences in *subjective* assessments of difficulty may nevertheless contribute to the extent of neural preparation, even if *objective* performance differences do not seem to.”

Each reviewer also provided some additional concerns and suggestions, which can be found in their specific individual reviews. I hope that these are helpful to you as you prepare your revision.Reviewer #1:[…]1) The specific memory-guided attention manipulation was unorthodox. In previous studies, learned associations guided spatial attention on the basis of the content of memories. Here, learned associations are used to signal the dimension to which participants must attend (art or room), without predicting any of the contents to be anticipated. This is like learning the meaning of an explicit attention cues directing individuals to switch or stay with their current attentional focus (as in studies by Yantis).In addition, the manipulation brings with it an element of endogenous, self-directed choice of what to attend supported by relatively short-term memories of what they encountered on a previous trial. The manipulation brings significant additional demands to the task – e.g., WM, multiplexing task at hand and goal setting, choice. A similar manipulation in spatial perceptual attention tasks was performed by Taylor, Rushworth and Nobre, 2008.The authors should consider and discuss how the particularities of their design will have affected their findings relative to how memory-guided attention has been studied in the past.

These are great papers and incredibly relevant. Thank you for pointing them out. We agree that our manipulation is unorthodox and that there are a number of key differences between our paradigm and other studies of memory-guided attention. We now mention these in the Discussion, in the subsection entitled “Relation to prior studies”:

“Our work was inspired by studies of memory-guided attention (e.g., Stokes et al., 2012; Summerfield et al., 2006) but it differs from them in a number of ways. […] Second, our study converges with other studies of memory-guided attention in suggesting that the hippocampus plays a role in guiding attentional behaviors on the basis of past experience (see Aly and Turk-Browne, 2017, for review).”

“We also found that these regions showed no difference in univariate activity levels between the memory-guided and explicitly instructed conditions during the orienting period. […] Thus, differences in the kind of information carried by memory (specific content vs. abstract task set), as well as in the timing of the orienting periods and the attention task, could have led to the observed differences in univariate activity during preparatory attention.”

2) The authors are interested in predictive anticipatory states, but there are no long periods during the task in which stimulation is absent for deriving clean anticipatory neural states. The 'decision period' is short and insufficiently separated from the image period.(Calling it a decision period is also confusing, since that would typically relate to the decision at the probe phase. Perhaps the 'orienting' period?)The authors address this issue (subsection “Robustness of preparatory attentional states”), but the limitation in the design means the issue remains inconclusive.

We agree that ‘decision period’ is not the best terminology. We have therefore replaced this term with ‘orienting period’, as recommended. More importantly, we agree that a study with a longer delay between the orienting period and the image period would be ideal, perhaps in conjunction with methods with higher temporal resolution than fMRI. We now mention this and other caveats and alternative explanations in the Discussion. The relevant text is pasted below, and we discuss further limitations in other responses to reviewer comments.

“The representational nature of the preparatory attentional states that are observed in the present study therefore deserves further investigation. […] This would be particularly useful if fMRI were complemented with EEG, to incorporate the high temporal resolution of the latter method (e.g., Stokes et al., 2012).”

“An alternative possibility is that the hippocampus is capable of preparing for upcoming attentional states equally strongly regardless of how these states are guided (i.e., by memories vs. explicit instructions) – but we were not able to observe this in our task because of limitations of the experimental design. […] Such methods can establish the temporal dynamics by which memory-guided vs. explicitly instructed attention influence representations across different brain regions.”

We hope that these new sections adequately describe the limitations of the current study, while laying out how future studies can be designed more optimally. Furthermore, we hope that the numerous control analyses and thorough consideration of autocorrelation (subsection “Robustness of preparatory attentional states”) help clarify the relative robustness of our results despite these caveats.

3) Given the additional demands of the memory task, could these have contributed to univariate differences in HC and mPFC activations between the task conditions? How can one rule this out?

We agree that this is a reasonable explanation. We tried to be careful not to over-interpret the univariate activity difference because, indeed, there are many potential reasons for it (also see comments by reviewer #2). For example, this difference might arise because of the demand to monitor the search set for stay/switch cues, the demand to retrieve the meaning of those cues, or from some other cognitive process arising from the dual-task nature of the memory-guided condition. We now mention this in the Discussion.

“For example, during the attentional search task (i.e., during the image period), hippocampus and vmPFC univariate activity levels were higher for memory-guided vs. explicitly instructed attention (Figure 3). […] Thus, many potential cognitive functions can account for the univariate activity enhancement in hippocampus and vmPFC during memory-guided attention in this study.”

We do note, however, that these univariate activity differences cannot be explained by differences in difficulty between the memory-guided and explicitly instructed tasks:

“If the hippocampus and vmPFC are more involved in attentional behaviors that are guided by memory, then they should show enhanced univariate activity during the memory-guided vs. explicitly instructed task. […] In the Discussion, we further consider what enhanced univariate activity in these regions might reflect.”

4) Given the nature of the attention manipulations, the multivariate effects were not linked to any content in the images, but to the dimension of the images (art, room) that was relevant. Could patterns also reflect nuisance factors other than focus on the information relevant to the goal? For example, some participants may have found one of the dimensions more difficult and therefore modulated arousal or effort levels to compensate and achieve similar levels of performance?

This is a good point and brings up the important issue that the representational content of the preparatory signals is unclear. By correlating orienting period activity patterns with image period activity patterns, we hoped to identify cognitive features of the art/room tasks that are “reinstated” in preparation for doing that task. However, this still leaves a great deal of flexibility in what is being reinstated: individuals could be bringing to mind an abstract attentional state (attend to global features vs. local features; attend to geometry vs. color), a task instruction (find a similar painting vs. find a similar room), or a metacognitive state (“The art task is harder for me, so I should expend more effort”). As long as these cognitive states are common between the image period and the orienting period, they may be components of the observed preparatory attentional states.

We therefore conducted an analysis to test the proposed interpretation (about differences in difficulty/arousal/effort). Our goal was to determine whether the extent to which a task was difficult for a participant correlated with the magnitude of neural preparation. To do this, we examined individual differences in the magnitude of preparatory attentional states and how these were related to differences in task performance (specifically, the absolute difference in A’ between the art and room attentional states). If preparatory attentional states reflect a nuisance variable such as anticipating a more difficult task, then individuals who show greater performance differences between the art and room attentional states should also show stronger preparatory attentional signals. However, we found no evidence for this interpretation. That said, it is possible that *subjective* assessments of task difficulty contributed to anticipatory signals, even if *objective* performance was not related to these signals. We therefore mention this and other interpretations in the Discussion (subsection “Nature of preparatory attentional states”).

“When a brain region prepares for, or anticipates, an upcoming task, what is being represented? […] The representational nature of the preparatory attentional states that are observed in the present study therefore deserves further investigation.”

5) Separating retrieval of a task set from preparation to its utilisation is conceptually very difficult and may not be possible in practise. Without some content-related expectation that is separate from the remembered/cued task set, this topic cannot really be addressed in the current experimental design. In previous studies (e.g., Stokes et al., 2012), memory for an association predicted the specific location of a target, so that differential levels of activity in visual cortex could be compared accordingly.(However, to some extent, this distinction may not always be useful, and one might instead consider how an act of retrieval can itself change the state of the system in a way that changes how it processes incoming information.)

Yes, we absolutely agree. We are not able to separate retrieval of a task set from preparation of its utilization – and it is an interesting question of how and when it might be possible to do so. We noted two places in which we might have been unclear about this: in the Results and Discussion.

First, in the Results, this concern might arise in the subsection entitled “Retrieval of past states or preparation for upcoming states?” Our goal here was to try to separate retrieval of the past attentional state from retrieval of, and preparation for, the upcoming one. But instead, we framed the analysis as addressing retrieval vs. preparation, which we agree is not what we really do. Our switch-trial-only analysis suggests that activity patterns during the orienting period of trial *N*+1 do not reflect retrieval of the attentional state on trial *N*. However, these activity patterns might reflect retrieval of the task set/attentional state that is necessary for trial *N*+1 and/or preparation for that state – and we cannot tell the difference between those two. And of course, we agree that the act of retrieving the upcoming task may change the hippocampus and other brain areas in such a way as to prepare them for processing features relevant for that task. We now clarify this as follows:

“These results therefore suggest that, during the memory-guided task, hippocampal activity patterns during the orienting period reflect preparation for the upcoming attentional state rather than retrieval of the preceding attentional state. […] We discuss the content of such preparatory signals in more detail in the Discussion.”

Second, in the initial submission, we discussed how our hippocampal results may index preparation for upcoming attentional states, while the Stokes et al., 2012 findings may reflect memory retrieval of learned target locations (which are then used to prepare other brain regions to guide attention). We agree that this may be too simplistic, given that retrieval and preparation may be intricately related. We have therefore modified that section as follows:

“In order to use memory to anticipate upcoming attentional goals, one must first retrieve the relevant memory and then use it to prepare for the upcoming task at hand. […] Future studies using methods with high temporal resolution (e.g., MEG/EEG) will be useful for determining the temporal dynamics by which the hippocampus switches from retrieving a past memory to using that memory to anticipate upcoming attentional states – if indeed, these are separable processes as opposed to inherently linked.”

6) Although the focus on HC and mPFC is well motivated and sensible, is there a risk that the authors miss where the real action is? It would be good to provide a supplementary exploratory analysis (using appropriately strict corrections for multiple comparisons) to reassure readers that the study captures the main players in in the cognitive functions examined and/or to highlight any additional important brain area/relationship that could be pursued in future studies.

Thank you for this suggestion, which we agree is important. Our main goal in this paper was to determine how the brain uses memories to prepare for upcoming attentional goals. Thus, we conducted a whole-brain searchlight analysis to find regions that showed greater preparatory coding for memory-guided vs. explicitly instructed attention. No voxels survived correction for multiple comparisons (p <.05 family-wise error correction, voxel-based thresholding). We also conducted whole-brain searchlight analyses for the memory-guided and explicitly instructed conditions separately. A few isolated voxels survived multiple comparisons correction, but no meaningful clusters emerged.

Of course, one cannot over-interpret these findings – the hippocampus and vmPFC did not appear in the whole-brain analyses at the corrected threshold, either. It is therefore possible that other brain areas are involved in preparing for upcoming attentional states, but not strongly enough to be seen with the strict multiple comparisons corrections. We now describe these results and the analysis approach in the manuscript and have included Figure 5—figure supplement 2, that depicts the isolated voxels that survived multiple comparisons correction.

First, in the Results, we now have a brief subsection entitled “Attentional preparation in other brain regions”:

“Although our focus has been on the hippocampus and vmPFC, we conducted exploratory whole-brain analyses to investigate neural signatures of attentional preparation elsewhere in the brain. […] These results must of course be treated with caution: it is very likely that brain areas other than the hippocampus and vmPFC prepare for upcoming attentional goals, but more targeted region-of-interest analyses are required to uncover them.”

Second, we describe the searchlight analysis in the Materials and methods, in a brief subsection entitled “Orienting Period – Whole-Brain Searchlight”:

“To test whether other brain regions represent preparatory attentional states, we performed the orienting period pattern similarity analysis using a whole-brain searchlight approach, via the Simitar toolbox (Pereira and Botvinick, 2013). […] Voxel-based thresholding was applied, corrected for multiple comparisons using the family-wise error rate correction (p < 0.05).”

Reviewer #2:[…]1) This was a thorough and well written report of a variation on the artist/room task. My first comment, probably apparent from my summary, is that I am not entirely on board with the whole framing around attention or "attentional states". The terminology has been used by the authors previously, but I find it somewhat idiosyncratic. The way I would write about this same experiment: there are two conditions, memory-guided vs. explicitly instructed. Participants are performing one of two tasks, artist or room. We are looking at how the task is represented when cued or being performed. Yes, the task determines what participants should be paying attention to, but that does not necessarily mean that what is being decoded by RSA is best described as "attentional state". Something like "task representation" seems more neutral, and it would align with terminology commonly used in task-switching literature that looks at very similar questions. Even in memory, we have studies that use living/non-living judgment on some trials and bigger/smaller than shoe box on other trials. They are still described as two tasks (not attentional states) and have been also analyzed similarly to look at task (or "task set") representation. I am not opposed for the authors to put forth their preferred interpretation in the Discussion. But I find the term "attentional state" over-interpretative and potentially misleading when used as the main framing. Also, it was never defined.

Thank you for pointing this out. We agree that we should have defined “attentional state”, and certainly should have brought up the links to task representations, task sets, and task switching. We now cite this work where appropriate and have made a number of changes to make our terminology more clear.

We agree that the type of attention we manipulated diverges from many other studies of attention. Such studies often manipulate attention to specific visual features (e.g., a particular line orientation, spatial location, or color) rather than higher-level dimensions (e.g., spatial geometry, artistic style). We now discuss this in a lot more detail. Furthermore, we highlight that – although unorthodox – our attentional manipulation did produce robust behavioral findings that are hallmarks of an attentional manipulation (better performance on validly cued trials vs. invalidly cued trials). The changes we made are highlighted below. We agree that there are many potential ways to frame this paper and hope that our rationale is now clearer.

In the Introduction, we now define “attentional state”:

“Here, we examine the mechanisms underlying memory-guided attention with the aim of determining the nature of neural representations that enable past experiences to be used to prepare for upcoming attentional states. […] Attentional states can be considered an instance of a task representation or a task set (Mayr and Kliegl, 2000; Sakai, 2008), with the task defining what should be attended to.”

In the Discussion and Materials and methods, we go over the similarities and differences between our manipulation and those used in other studies of memory-guided attention and bring up similarities to the task-switching literature.

“Our work was inspired by studies of memory-guided attention (e.g., Stokes et al., 2012; Summerfield et al., 2006) but it differs from them in a number of ways. […] Second, our study converges with other studies of memory-guided attention in suggesting that the hippocampus plays a role in guiding attentional behaviors on the basis of past experience (see Aly and Turk-Browne, 2017, for review).”

“After a key was pressed on the initiation screen in the explicitly instructed task, the attentional cue (“ART” or “ROOM”) was randomly assigned. In the memory-guided task, participants were instructed to select their attentional state based on the stay/switch cue in the preceding trial. […] For example, if the attentional state on the previous trial was “art”, and there was an art “switch” cue, then the attentional state on the current trial should be “room” (art stay/switch cues only appeared on trials where art was attended; room stay/switch cues only appeared on trials where rooms were attended).”

2) Univariate analyses (Figure 3): More control needs be done for task/condition difficulty differences. Memory vs. external condition behavioral performance difference was p = 0.084 (subsection “Behavior”), memory vs. external condition hippocampus effect was p = 0.011. As the former-behavior-was dismissed as no difference, it was not considered as a possible source of the latter. However, control analyses should be conducted, taking the marginal behavioral differences into account. This holds for results from both Figure 3A and Figure 3B.

Yes, thank you for pointing this out. We absolutely agree. We will note that only valid trials were used in the univariate analyses shown in Figure 3A and Figure 3B (in keeping with our prior work, and to ensure that the results are not due to ‘contamination’ from invalid probes). The complementary behavioral analysis, which includes valid trials only, was associated with *p* = 0.20 for the difference between conditions. Nevertheless, your point still holds, and we have therefore conducted these analyses. All of the fMRI results in Figure 3 hold after controlling for differences in behavioral performance, as discussed below.

“If the hippocampus and vmPFC are more involved in attentional behaviors that are guided by memory, then they should show enhanced univariate activity during the memory-guided vs. explicitly instructed task. […] In the Discussion, we further consider what enhanced univariate activity in these regions might reflect.”

3) Univariate results interpretation. Even if activation difference in Figure 3 hold after controlling for behavioral differences, what they mean may be different from what is proposed. The results are from the image period, where "attentional state" (room or artist task) is no longer memory-guided. The “ART” or “ROOM” cue was explicitly presented once the memory-guided decision was made earlier in the trial. By the image presentation period, the cue became external in both conditions, making them comparable in that regard.What differentiates the conditions instead is that the memory-guided condition is a dual-task condition. Participants need to conduct the room or art task AND ALSO keep track of a potential switch/stay cues (that needed to be memorized). This offers a simple explanation of any activation differences between the memory-guided and explicitly instructed condition.

We absolutely agree that this is a potential explanation, and reviewer #1 also made a similar point. We now mention this explanation in the Discussion:

“For example, during the attentional search task (i.e., during the image period), hippocampus and vmPFC univariate activity levels were higher for memory-guided vs. explicitly instructed attention (Figure 3). […] Thus, many potential cognitive functions can account for the univariate activity enhancement in hippocampus and vmPFC during memory-guided attention in this study.”

4) Operationalization of the "decision period". I did not have any issues with the RSA analysis per se, and appreciated the authors' thorough control analyses. However, I did not find it appropriate to call the beginning of the trial as "decision period" for two reasons. (1). There wasn't a task-relevant decision to be made in the explicitly-instructed period. (2) The art/room cue period was included. Given their proximity, there is no way to get rid of the art/room cue-related activation from this analysis. My view of this analysis and the results is that we are primarily measuring cue-related task representation (room/art). For example, both hippocampus and vmPFC represent “ROOM” vs. “ART” above chance in the explicitly instructed condition, which clearly does not come from the decision whether to press index or middle finger. Thus, the term "decision period", which lends itself to implying decision-related activity, is misleading in the current context.

We agree that ‘decision period’ is not the best terminology, and reviewer #1 pointed this out as well. We have therefore changed it to ‘orienting period’, following the advice of reviewer #1.

5a) Additional concerns related to the interpretation of the multivariate results: As noted above, most straightforward interpretation of Figure 5 does not involve "decision" per se. It seems to be about room/art task set representation ("which task I'm doing"?). The match between the cue period and image period indicates that the room/art task representation is abstract in nature (not the specific task cues "ART" or "ROOM"). Whether it is appropriate to call that preparatory attentional signal is not apparent. Of course, participants need to remember which task they are doing throughout the image period. Thus, one may also see the relationship the other way – decoding during image period reflects memory for the cue from the beginning of the trial.

We agree that observing similar activity patterns during the orienting period and the image period does not in itself specify the content of these representations. reviewer #1 raised a similar point as well, so we have considerably revised the interpretation offered in the Discussion:

“When a brain region prepares for, or anticipates, an upcoming task, what is being represented? […] The representational nature of the preparatory attentional states that are observed in the present study therefore deserves further investigation.”

5b). In addition, the authors found stronger representation in memory-guided than explicitly instructed condition in the hippocampus. A challenge with interpreting this result is different duration of the event of interest under the two conditions. In the memory-guided condition, it is already apparent during the decision screen (and on the majority of trials, even before the decision screen during ITI) whether the next trial is the room task or the art task. Thus, brain activation during both decision and task-cue screens (modeled together here) can reflect the room/art task identity. In contrast, in explicit condition, the room vs. art task information is not available until the task-cue screen, and thus the brain cannot possibly reflect room vs. art task identity while the subject is deciding whether they want to press their index or middle finger. Perhaps I misunderstood the procedures. But if not, I don't see the differences between art vs. room decoding between the conditions as interpretable.

We agree that the upcoming task is known for different lengths of time in the memory-guided vs. explicitly instructed conditions. As the reviewer noted above, though, it is critical that the attentional cue (“ART” or “ROOM”) was included in the orienting period for the fMRI analyses – otherwise there is no possible way to observe preparatory task representations in the brain for the explicitly instructed condition. Thus, including the attentional cue made the comparison between conditions fairer.

However, if we were only ever able to measure preparatory task representations in the memory-guided condition, one might worry that the attentional task was simply not known for long enough to yield a reliable preparatory signal in the explicitly instructed condition. But vmPFC showed equally strong preparatory signals for both conditions, suggesting that this is not the case.

That said, it is certainly worth discussing this issue in detail, because the conditions differ not only in the source of information about the upcoming task (memory vs. explicit instruction) but also for how long that information is known. This is a limitation that we discuss as follows:

“An alternative possibility is that the hippocampus is capable of preparing for upcoming attentional states equally strongly regardless of how these states are guided (i.e., by memories vs. explicit instructions) – but we were not able to observe this in our task because of limitations of the experimental design. […] Such methods can establish the temporal dynamics by which memory-guided vs. explicitly instructed attention influence representations across different brain regions.”